

## Causes of uncertainty in observed and projected heterotrophic respiration from Earth System Models

Cary Lynch[1], Corinne Hartin[1], Min Chen[1], and Ben Bond-Lamberty[1]

[1]Pacific Northwest National Laboratory, Joint Global Change Research Institute, 5825 University Research Court, Suite 3500, College Park, MD 20740

*Correspondence to*: Cary Lynch (cary.lynch@pnnl.gov)

### Abstract

Heterotrophic respiration (RH) is a large component of the terrestrial carbon cycle, but one poorly simulated by Earth system models (ESMs), which diverge significantly in their historical and future RH projections. There is little understanding, however, of the causes of this variability and its consequences for future model development and scenario evaluation, and examining the relationships between RH and key climate variables may help to understand where and why models are divergent. We quantified the statistical relationships between RH and other terrestrial/climate variables across a suite of 25 ESMs from the Coupled Model Intercomparison Project phase 5 (CMIP5) for the 20[th] and 21[st] centuries, comparing the models both to each other and to an observation-driven global RH dataset. Compared to observations, ESMs consistency overestimate both the magnitude and climate sensitivity of global RH. The relationship between RH and surface air temperature (TAS) is strong, especially at high latitudes, and largely consistent across models. The observed RH and precipitation (PR) relationship is strong and positive ($r \geq 0.5$, $P < 0.005$), but few models consistently show this sensitivity of RH to PR. The RH-TAS relationship explored here, and more pattern scaling methods more generally, can be used to efficiently explore uncertainty and projected changes in RH under a wide range of future emission scenarios, and understand how models' structural and parametric choices produce divergent results. Because uncertainty in RH has large effects on ESM projections of future climate, this may help direct attention to relationships in the carbon cycle that contribute to this uncertainty.

## 1 Introduction

Soil heterotrophic respiration (RH), the soil-to-atmosphere $CO_2$ flux derived from microorganisms' metabolism of litter detritus and organic carbon, constitutes a large and highly uncertain component of the terrestrial carbon cycle (Hashimoto, 2012; Hashimoto et al., 2015; Luo and Zhou, 2006). This carbon flux may result in a significant climate feedback in the future, as mineralization of long-stored soil carbon releases C to the atmosphere (Bond-Lamberty and Thomson, 2010b; Friedlingstein et al., 2014). This will be dependent on how strongly large-scale processes, including RH, are affected by abiotic drivers such as temperature and precipitation (Bond-Lamberty and Thomson, 2010b; Hursh et al., 2016; Sierra et al., 2015). While both temperature and precipitation have a positive effect on the global terrestrial carbon flux (Li et al., 2017; Liu et al., 2016), the effect of these drivers can change in complex ways depending on the spatial



scale being examined (Jung et al., 2017).

Earth System Models (ESMs) generally project increases in the RH flux due to global climate change (Friedlingstein et al., 2014). However, RH model structure and
parameterizations are quite simple relative to many other processes (Wieder et al., 2013): RH is typically treated as a first-order decay process occurring in litter and distinct soil layers, with remaining leaf litter and woody debris ultimately transported to the belowground carbon pool after fire losses (Shao et al., 2013). As a result, ESMs generally do not accurately capture either the observed RH flux (Shao et al., 2013) or the spatial
distribution of soil carbon (Todd-Brown et al., 2013). At the global scale net primary production (NPP) and RH are of roughly equal magnitude (Hashimoto, 2012; Zhao and Running, 2010), and interestingly, ESMs appear to better capture the observed NPP flux (Todd-Brown et al., 2013). This discrepancy between NPP and RH would suggest that there is large model uncertainty in RH quantification, and this uncertainty can have large
effects on ESM predictions of the 21[st] century Earth system (Friedlingstein et al., 2014).

Thus, it is important to understand and explore the uncertainty in RH quantification. Pattern scaling, one technique for doing so, can be used to examine key relationships between RH and abiotic drivers such as surface air temperature (TAS), as well as carbon
fluxes such as NPP. In pattern scaling, a 'pattern' is the relationship between a local and global variable, and is intended to provide a measure of local sensitivity to global change (Mitchell, 2003; Osborn, 2009; Santer et al., 1990). These patterns can then be scaled in magnitude by a specified global mean RH, NPP, or TAS change obtained from a simple climate model (SCM). It is a simple, flexible approach that allows for a computationally
efficient analysis of a wide range of future scenarios that have not been simulated by ESMs. Pattern scaling may help to understand why certain models show a larger future trend than others by examining the relationships between RH and other variables, and comparing model output to observed data.

In this study, we analyzed historical RH from an ensemble of ESMs and compared these outputs to an observation-based data product. Then we examined the statistical uncertainty of projected RH across models and projections, and explored the spatial and temporal relationships between RH and other variables. Finally, we investigated whether pattern scaling can be used to better quantify changes and uncertainty in projected RH,
and understand the characteristics, strengths, and weaknesses of different ESMs with respect to RH and the terrestrial carbon cycle.

## 2 Methods

*2.1 Datasets*

Table 1 lists the observed datasets used to evaluate ESM RH outputs. The Hashimoto et al. (2015) soil respiration (RS) data include an annual gridded RH product created by assimilating observed RS into a statistical model to achieve continuous spatial and
temporal coverage for the 20[th] and early 21[st] centuries (Hashimoto et al., 2015);(Hashimoto et al., 2015)RH is then computed based on a simple empirical



relationship (Bond-Lamberty et al., 2004). RH changes over time in this dataset, but these changes are driven entirely by climate and do not include process- or land-use based changes in the carbon cycle. Currently, because RH cannot be directly measured at scales larger than a few square meters (Bond-Lamberty et al., 2016), the Hashimoto et al.

(2015) dataset is the best available estimate of both the spatial distribution and global total RH flux (Xu and Shang, 2016).

Other observation/reanalysis datasets used include monthly NCEP TAS (Kalnay et al., 1996) and NASA precipitation flux (PR; Adler et al., 2003) reanalysis products; and an

annual NPP, satellite derived product (Zhao et al., 2005).

We used model output from two sets of experiments from the Coupled Model Intercomparison Project Phase 5 (CMIP5; Taylor et al., 2012). The 'historical' experiment was used to evaluate model performance metric as compared to observations.

Model historical runs varied in length, so we used 1901 as the start of the historical period, and 2005 as the end. For future projections, we used the high-forcing RCP 8.5 scenario, in which radiative forcing increases to 8.5 W/m$^2$ in 2100 (Riahi et al., 2011). For the future simulations, the start year was 2006, and the end year was 2099.

For the assessment of patterns, we used all available climate models, resulting in an ensemble of 25 ESMs (Table 2), and we only used the first realization (i.e., ensemble member) from each model. No performance weights were constructed, and the assumption of model independence was implicit.

Most of the analysis was limited to the area between 80°N and 60°S. All observed and model output was regridded to the lowest spatial resolution (2.8° by 2.8°) of the multi-model ensemble prior to the calculation of an ensemble mean. This was done for averaging purposes, as each model had a different spatial resolution. Regridding to the lowest resolution of the multi-model ensemble is a conservative assumption that avoids

interpolation errors.

*2.1.1 Pattern scaling*

Pattern scaling (Santer et al., 1990) was used to estimate statistical relationships between

local (subscript L) and global mean/total (subscript G) climate variables. Here, a least square regression (LSR) approach was used (Kravitz et al., 2017; Lynch et al., 2017) in which patterns were calculated from the RCP 8.5 future forcing scenario for all models as:

$Y_L = \alpha + \beta * X_G + \varepsilon$

In this equation, $X_G$ is the global annual mean (TAS) or total (RH, PR, NPP) climate predictor (one-dimensional, unsmoothed), and $Y_L$ is the gridded local climate dependent variable (three dimensional). $\beta$ is a two-dimensional field of regression slopes, and $\varepsilon$ is a

three-dimensional residual term (error) stemming from linearly fitting the dependent variable to the predictor. $\alpha$ is the y-intercept, which we take to be 0 by only computing





change, not absolute values. Calculated patterns are described in terms of 'sensitivity', i.e. amount of local, grid-cell change per 1 unit global change.

To evaluate pattern accuracy, we quantified the differences between the reconstruction $\hat{B}$ and the actual model output B via the root mean square error (RMSE) over the area-weighted difference at the end of the 21st century. In this instance RMSE is used to describe how well the predicted pattern emulates the actual model change, with lower RMSE indicating that the predicted pattern better captures the actual model change.

$$RMSE = \frac{\sqrt{\sum_x \left[ \left( \hat{B}(x) - B(x) \right) \cdot A(x) \right]^2}}{\sqrt{\sum_x \left[ A(x) \right]^2}}$$

where $A(x)$ is the area of the grid box $x$ and sums were calculated over all $x$.

**3 Results**

*3.1 Observed trends and relationships*

Most models do not capture the magnitude or trend of observed annual global RH (Figure 1, and Supplementary Figure 1). The 25 CMIP5 models examined exhibited increasing
RH throughout the historical and future periods, from a mean of 60 (range of 45-83) Pg C yr$^{-1}$ in 1901 to 90 (range of 57-139) Pg C yr$^{-1}$ by the end of the 21st century. In contrast, observed data support a best estimate of 51 Pg C yr$^{-1}$ in 1901 and only 52 Pg C yr$^{-1}$ in 2013. Most models thus overestimate, by about 8 Pg C yr$^{-1}$, global RH as compared to our best statistical upscaling estimate of global RH. In the observed period the
observations do not have a statistically significant trend (0.005 Pg C yr$^{-1}$ for 1901-2010), but the majority of the climate models have a slight positive trend (0.02 to 0.1 Pg C yr$^{-1}$ for 1901-2005).

Under the RCP 8.5 forcing scenario, the projected trend is strongly positive (0.06 to 0.6
Pg C yr$^{-1}$ for the 21st century) and the model spread is larger than it is in the historical period. The HadGEM models project the largest 21st century change in global RH, while the NorESM models project the smallest change (Supplementary Figure 1), which is consistent with NPP results from Todd-Brown et al (2014).

Models vary in their estimation of interannual standard deviation (Figure 2). Even though the historical interannual standard deviation is relatively large for some models, it stays constant for much of the 20th century. Models from the same modeling center are very similar in their projected trends in standard deviation. In the future, the interannual standard deviation of RH is projected to change from a mean of 0.98 Pg C in 1900 to 1.35
Pg C in 2100 (Figure 2). The majority (65%) of the models show an increase in interannual variability, but some project standard deviation to stay the same (CCSM4/CESM) or even decrease (GISS-E2 and NorESM) by the last half of the 21st century. The models with the lowest (or decreasing) interannual standard deviation are



also the models with the smallest projected trend (Supplementary Figure 1).

The individual models exhibit significant differences in correlation, bias, and error (Figure 3). The majority of the models in the ensemble had a positive bias, and the average positive bias was larger than the negative bias. The MPI models (22-23) had the largest positive bias and the CCSM4-CESM models (5-8) had the largest negative bias. The MIROC models (20-21) had the smallest bias overall. The MPI models (22-23) had the largest spatial correlation values, and the IPSL models (17-19) had the smallest spatial correlation values. The GISS-E2 models (11-14) had the lowest RMS differences, while the HadGEM models (15-16) had the highest RMS differences. For both simple metrics of historical performance, models from the same modeling center had similar bias and spatial correlation magnitudes.

The modeled and observed relationship between RH and surface temperature (TAS) exhibit latitudinal patterns (Figure 4a and Supplementary Figure 2a). Correlation values are greater at high latitudes, presumably because the dominant control on RH is temperature in these cold biomes. At lower latitudes, the relationship between RH and TAS is weaker, presumably due to other controls on RH such as precipitation and soil moisture. In general, there are strong negative relationships between RH and TAS where soil-moisture and precipitation are limited (cf. Shao et al. 2013).

Few models captured this observed spatial correlation (black line): the CCSM/CESM models (green lines) captured the temporal autocorrelation patterns the best and the MPI and BCC models (cyan and red lines, respectively) performed the worst. Most models tended to perform better in the Northern Hemisphere mid-latitudes, and less well in the Northern Hemisphere sub-tropics where the observed relationship is largely negative. In addition, the model spread of the RH-TAS relationship becomes progressively larger from the Northern Hemisphere to the Southern Hemisphere. This is likely due to less land (and thus higher variability in the model averages) in the Southern Hemisphere, as well as generally fewer RH observations from the Southern Hemisphere more generally (Epule, 2015).

We performed a parallel analysis for RH and PR. The observed $r$ values are strong and positive ($r \geq 0.5$, $P < 0.005$, Figure 4b and Supplementary Figure 2b), strongest at mid and high latitudes, and weakest in the tropics (20°N to 20°S) along subtropical dry zones. These are also regions where the direction of the relationship is spatially heterogeneous (Supplementary Figure 2b). The modeled relationship between historical RH-PR is stronger than the relationship between RH-TAS, but unlike the RH-TAS relationship, the RH-PR relationship do not exhibit consistent geographical patterns. Interestingly, the models that overestimated the RH-TAS $r$ values (Figure 4a), are the same models that underestimate the RH-PR $r$ values (Figure 4b).

The observed relationship between RH and NPP is weak (Figure 4c, Supplementary Figure 2c). However, some models have a strong positive relationship (HadGEM models), while others have a strong negative relationship (CCSM4/CESM and NorESM models). The observed strength and direction of the relationship between RH and NPP





also does not have any spatial heterogeneity.

*3.2 Projected relationships*

The projected strength of the local-global RH relationship is shown in Figure 5. The
diversity of histogram shapes highlights the ensemble spread, and models from the same
modeling center have very similar shapes. The models are grouped into strong (average
$R^2 \geq 0.75$; bcc, CanESM2, and IPSL models), and weak (average $R^2 \leq 0.25$;
CCSM4/CESM, MIROC, and NorESM models) RH local ($RH_L$) and RH global ($RH_G$)
relationships. The GISS-E2 models were separated and emphasized because the local-
global relationship diverged greatly from all other models.

Models with a small $R^2$ between RH local ($RH_L$) and RH global ($RH_G$) were generally the
most sensitive to changes in $RH_G$, particularly in the tropics and Southern Hemisphere
(Figure 6a), but there was not a clear difference between the strong and weak $R^2$ models.
However, the models with a smaller $R^2$ had the smallest 21[st] century trends
(Supplementary Table 1), weak/decreasing year to year standard deviation (Figure 2), and
an overestimation of the local-global patterns. Patterns were the least sensitive, i.e. had
comparatively small local-global change ratios, at high latitudes and in the tropics. The
groupings also emphasize the difference in projected trends for RH and NPP (Figure 6)
and those with a small $R^2$ have smaller projected trends than those with a large $R^2$.

The $RH_L$-$TAS_G$ relationship is different from the $RH_L$-$RH_G$ relationship, as there are
clearer differences between the stronger and weaker models (Figure 7, 6a-b). For the
models with a stronger $RH_L$-$TAS_G$ relationship, RH is much more sensitive (i.e., the
local-global change ratio is larger) to changes in global TAS everywhere (Figure 6b).
The relationships and patterns between $RH_L$-$PR_G$ (not shown) are very similar to the $RH_L$-
$TAS_G$ relationships. The GISS-E2 models were generally similar to the models with a
weak $RH_L$-$TAS_G$ relationship, but, like in Figure 6a, there are clear disparities at the
equator and in the Southern Hemisphere (Figure 6b).

The modeled $R^2$ between $NPP_L$-$NPP_G$ stronger and more consistent across models than it
is for the $RH_L$-$RH_G$ (Figure 8). For some models (bcc, BNU, CanESM, GISS-E2, and
HadGEM models), the $NPP_L$-$NPP_G$ $R^2$ are the inverse of the local-global $R^2$ for RH. The
$NPP_L$-$NPP_G$ spatial patterns are similar to the $RH_L$-$RH_G$, but the $NPP_L$-$NPP_G$ sensitivity is
stronger around the equator (Figure 6c). Models with a weak $RH_L$-$RH_G$ relationship are
more sensitive to changes in global NPP. Once again, the GISS-E2 models show very
strong local sensitivity to global changes along the equator and in the Southern
Hemisphere. The relationships and patterns between local-global NPP are highly similar
to $NPP_L$-$TAS_G$ (Figure 9, Figure 6d) and $NPP_L$-$PR_G$ (not shown), but NPP in the GISS-E2
models is highly sensitive to changes in global temperature.




For the future, the $RH_L$-$TAS_G$ pattern has the best fit as inferred from very low standardized error (Figure 10). This is likely due to the clear and strong global mean temperature signal projected from the RCP 8.5 scenario. The $RH_L$-$NPP_G$ pattern standard error is also low, and similar spatial features of the pattern standard error are evident

across all four patterns. Larger errors are evident in the tropics and generally are larger along leeward/eastern coast of most continents.

RMSE values between actual and pattern predicted RH were small, with the lowest RMSE when using the $RH_L$-$TAS_G$ (Table 3). RMSE were smallest for models that had a

weak $RH_L$-$TAS_G$ relationship, particularly the CCSM4/CESM, NorESM, and GISS-E2 models. Inversely, RMSE were largest for models with a strong $RH_L$-$TAS_G$ relationship. Low RMSE errors between the actual and predicted RH is likely due to clear and significant global trends (Supplementary Table 1) and the strongly linear local-global relationship between RH and RH/TAS/NPP, which was also shown in Figure 10.

## 4 Discussion

### 4.1 Observed trends and relationships, model comparison

In the observed period, both the magnitude and change in global RH are overestimated by the ESMs examined here. Models in the ensemble do not capture the observed trends and magnitudes in RH well, and there is a large model spread in the observed and projected relationship. On one hand, this is unsurprising, as ESMs tend to overestimate the effect strength of $CO_2$ fertilization on other parts of the carbon cycle such as NPP (Smith et al.,

2015). On the other, however, the observed RH dataset used in this study may underestimate the real trend due to the simple empirical relationship between soil respiration and RH used to construct the data (Hashimoto et al., 2015). With these caveats in mind, the NorESM and CCSM4/CESM models generally performed best when comparing model performance to observations. These models have identical number of

litter and soil pools, temperature and moisture functions, and representation of nitrogen cycling (Todd-Brown et al. 2013, Table 3). In addition, the NorESM and CCSM4/CESM models strongly underestimate total soil carbon, NPP flux, and soil carbon turnover time (Todd-Brown et al. 2013, Figure 2; Hashimoto et al. 2015, Figure 9).

Despite a large spread in estimated annual RH, the CMIP5 models robustly project a statistically significant positive trend in global RH and increased interannual variability under RCP 8.5. The notable exception to the projected increase in interannual variability are the GISS-E2 models, which have no carbon litter pools (Todd-Brown et al., 2013), but a large number (9) of soil pools. Also, temperature and moisture functions of soil

carbon estimates in the GISS-E2 models are strongly linear with no upper limit, and there is no nitrogen cycling. For the future period, the GISS-E2 models were also notable in that they project large increases in NPP, but comparatively small trends in RH. The seemingly too-high sensitivity of model RH to changing climate is very tentative, given the limitations on RH observations and available upscaled products (Bond-Lamberty and

Thomson, 2010a; Hashimoto et al., 2015; Shao et al., 2013), but if confirmed has significant implications for the assessment of these land models' performance and ability



to predict future climate change (Friedlingstein et al., 2014; Sitch et al., 2015).

Interestingly, model skill in simulating RH does not necessarily correspond to skill in other parts of the carbon cycle. For example, the MIROC and MPI models generally

performed the best in simulating RH, but these models largely overestimate soil carbon, NPP, and carbon soil turnover time (Figure 2 in Todd-Brown et al. 2013). The NorESM models in Figure 3 (models 24/25) performed well compared to observations, but in Todd-Brown et al. (Table 3, 2013), the NorESM models had very low Taylor scores across empirical datasets. The reverse is generally true with the HadGEM and IPSL

models: Table 3 in Todd-Brown et al. (2013) suggests that the HadGEM2 and IPSL models performed well in capturing observed RH characteristics, given that they performed well with respect to soil C turnover time, but these models had large RMSE (error) values in our results (Figure 3). These discrepancies could be due a number of reasons: (i) high and divergent spatial variability in certain models, and between models

and observations, in the abiotic drivers of RH (ii) an underestimation of global annual RH values in the observed dataset (Hashimoto et al. 2015); and (iii) the fact that models can perform well in reproducing soil C pools without doing so for fluxes, or vice versa, due to the first-order input and output (i.e., RH) algorithms used (Carvalhais et al., 2010). Robustly constraining and evaluating the ability of ESMs to simulate soil processes will

require jointly evaluating their pools as well as fluxes. In this regard, our work extends and complements the results of Todd-Brown et al. (2013).

### *4.2 Drivers of RH: observations and models*

Because respiration is strongly affected by climate, there is a strong zonal response in observed RH. This is evident in observations, but not in the ESMs. Robust agreement in strength and sign of the RH-TAS relationship is limited to Northern Hemisphere high latitude cold biomes, where TAS is generally considered the dominant control on the soil-to-atmosphere $CO_2$ flux (McGuire et al., 2009). As noted in the results, the models that

overestimate the RH-TAS relationship are also the models that underestimate the RH-PR relationship, and inversely those that underestimate RH-TAS overestimate RH-PR. In general, there are strong negative relationships between RH and TAS where soil-moisture and precipitation are limited (cf. Shao et al. 2013). This may be a reason why there is such a large spread values/trend in modeled RH.


The observed RH-PR relationship is strong, but the models strongly underestimate the observed RH-PR relationship. The observed zonal features of the RH-PR relationship are also not captured by the models. Given that the models do poorly in capturing the observed relationship, it is not surprising that Table 4 from Todd-Brown et al. 2013,

indicates that by including a soil moisture term, the models do not perform better in capturing NPP (instead, they found that modeled NPP is largely constrained by TAS). Some of this discrepancy may be due to the complicated scaling dynamics that occur in carbon-cycle drivers (Jung et al., 2017). Another reason for this inconsistency may in part be due to robust overestimation in seasonal and annual precipitation by the CMIP5

models (Liu et al., 2014). In semi-arid regions, high inter-model precipitation variability is also a problem.



The observed relationship between RH and NPP is not strong. While at the global scale the NPP (Zhao and Running, 2010) and RH (Hashimoto et al., 2015) carbon fluxes are of similar magnitude, and in theory should be equal in steady-state ecosystems (Chapin et

al., 2006), this was not true in either the models or observations examined here (Figure 6). There are many possible reasons for this: lag effects between climate-driven NPP changes and RH equilibrium (Baldocchi et al., 2006; Zhou et al., 2010); lag effects from past disturbances (Harmon et al., 2011); and poorly measured storage terms. The weak observed RH-NPP relationship may also be partially due to the short time span used

(2000-2013) in calculating the observed relationship, which both increases error and is short relative to SOC turnover times (Todd-Brown et al., 2013). Another potential problem with the observed datasets used may be that we are comparing a bottom-up statistical product (RH) with a top-down satellite/model one (NPP), with inevitable spatial and methodological mismatches. Moreover, studies disagree in strength and

direction of observed NPP trends. From satellite based data the trend is slightly negative (Zhao and Running, 2010), but from modeled studies which assimilate observed data (Ahlström et al., 2012; Sitch et al., 2015), the trend is strongly positive.

The modeled RH-NPP relationship is much stronger than the observed relationship. The

notable exception to this is the CCSM4/CESM models (green lines in Figures 4, 5, and 6) which have a negative RH-NPP relationship in the tropics and in the Southern Hemisphere. However, the model spread is quite large, with no notable zonal features. This may indicate that the models are too closely coupled, i.e. that their production and decomposition functions are too tightly tied to each other (Hashimoto et al., 2015), or

perhaps that the modeled parameterizations in RH/NPP quantification are too simple to capture necessary feedbacks in the carbon cycle. Theoretically at least, an RH-NPP relationship is expected at some scale, so it is difficult to diagnose whether the issue is with the observations used or model parameterizations that couple the RH-NPP flux. Furthermore, most models overestimate observed NPP trends and values (Todd-Brown et

al. 2013), which may contribute to stronger than observed RH-NPP $r$ values.

Overall, modeled RH sensitivity to climatic factors seems unlikely to be correct (Zhou et al. 2009; Shao et al. 2013; (Shao et al., 2013; Sierra et al., 2015; Smith et al., 2015)). At high Northern latitudes, the models underestimate both the temperature and precipitation

(as a proxy for soil moisture) relationships compared to the Hashimoto et al. (2015) observationally-based dataset. This may be because it has been shown that carbon system model decomposition rates have a low sensitivity to climate factors in cold regions with limited moisture (Sierra et al., 2015). The soil respiration temperature response ($Q_{10}$) range in Earth system models is large, but is generally stronger than that

derived from observational studies (Beer et al., 2010; Bond-Lamberty and Thomson, 2010b; 2015), and ESMs in general exhibit strong carbon cycle responses to climate change (Anav et al., 2013, 2015; Ito et al., 2017). This likely contributes to the large model spread in RH. The common practice of using a global constant $Q_{10}$ term rather than a spatially heterogeneous $Q_{10}$ may also contribute to this over/underestimation (Zhou

et al., 2009).



*4.3 Evaluation and discussion of projected RH and pattern scaling*

Despite large differences in local-global RH $R^2$ values, the local sensitivity to global change ("patterns") is small and largely consistent across models. Given the large

ensemble spread in projected global RH, it is interesting that the RH-RH patterns are so similar, with the largest differences at the equator. Models with large $RH_L$-$RH_G$ $R^2$ values are also more sensitive to global TAS change. TAS (and NPP) is better parameterized in the ESM models than RH, and as such is likely to yield a more robust response in local RH with stronger confidence in the resulting pattern. Todd-Brown et al.

(2013 and 2014) found that NPP and soil carbon are strongly tied to temperature. This relationship is also evident in the observed data (Hashimoto et al. 2015) and has great potential for use in pattern scaling studies to examine alternative forcing scenarios.

As shown in Table 3, the $RH_L$-$TAS_G$, and to a lesser extent, the $RH_L$-$NPP_G$, relationship

can be used to reproduce spatial RH features with comparatively low RMSE values. This raises the possibility that pattern scaling techniques, which are straightforward to implement and computationally inexpensive, could be used as a routine diagnostic for ESM outputs. In particular, these techniques may be valuable with respect to RH, a flux for which so few observational data exist and the model-measurement spatial mismatch is

extremely large (Bond-Lamberty et al., 2016; Zhou et al., 2009).

Model comparisons in the earth sciences need to be performed carefully with a full understanding of their limitations (Oreskes et al., 1994). Nonetheless, it is important to note that in this analysis the GISS-E2 models (Schmidt et al., 2014) are significant

outliers in both their projected local-global RH relationships and temperature sensitivity. Todd-Brown et al (2013) noted that these models exhibited a noticeable discrepancy between NPP and soil carbon, which was partially attributed to the allocation of plant biomass in the litter model. The GISS-E2 models also are characterized by very low equilibrium climate sensitivity, transient climate response, and GHG-attributable

warming (Gillett et al., 2013), and projected changes in terrestrial carbon and surface temperature are not likely to be strongly coupled.

Finally, it is important to note that like any analytical technique, pattern scaling has both strengths and weaknesses. Patterns can be used to examine model differences in response

to particular forcings or trends in global climate parameters, diagnose problem relationships, explore future emission sensitivity, and show where relationships are weak or non-linear. However, pattern scaling, as a statistical emulation method, is strongly tied to the assumption of stationarity (Mitchell, 2003). Patterns derived from emission scenarios with strong mitigation are less accurate and prone to large estimation errors

(Tebaldi and Arblaster, 2014). Finally, if the projected trend in RH and/or the predictor variable is weak, the pattern error term is relatively large, and pattern scaling should be used with caution.

**5 Acknowledgements**


This research is based on work supported by the US Department of Energy, Office of



Science, Integrated Assessment Research Program. The Pacific Northwest National Laboratory is operated for DOE by Battelle Memorial Institute under contract DE-AC05-76RL01830.

*Code and data availability*

Code used to complete this analysis was done primarily in NCL and will be available in a GitHub repository once the paper is accepted. All data used in this analysis are given in Tables 1 and 2 and are openly available to the public.

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





## 7 Tables and Figures

**Table 1.** List of datasets and their attributes used in this analysis.

| Variable | Spatial Resolution | Temporal Resolution | Source (version) |
|---|---|---|---|
| Heterotrophic Respiration (RH) | 0.5° x 0.5° | Annual, 1901-2013 | Soil Respiration Database (SRDB), Department of Energy (DOE) Pacific Northwest National Laboratory (PNNL) |
| Surface Air Temperature (TAS) | 2.5° x 2.5° | Monthly, 1948-2013 | National Centers for Environmental Prediction, National Center for Atmospheric Research (NCEP/NCAR) Reanalysis Project 1, National Oceanic and Atmospheric Administration (NOAA), Earth System Research Laboratory Physical Sciences Division |
| Precipitation Flux (PR) | 2.5° x 2.5° | Monthly, 1979-2013 | Global Precipitation Climatology Project, National Aeronautics and Space Administration (NASA) Goddard Space Flight Center |
| Net Primary Production (NPP) | 0.5° x 0.5° | Annual, 2000-2013 | NASA MODIS TERRA product ID: MOD17A3 |

**Table 2.** List of the CMIP5 models used in this analysis, with their respective spatial resolution and organization.

| Model | Spatial Resolution | Organization |
|---|---|---|
| bcc-csm1-1 | 2.8° x 2.8° | Beijing Climate Center, China Meteorological Administration, China |
| bcc-csm1-1-m | 1° x 1° | |
| BNU-ESM | 2.8° x 2.8° | College of Global Change and Earth System Science, Beijing Normal University, China |
| CanESM2 | 2.8° x 2.8° | Canadian Centre for Climate Modeling and |





| | | Analysis, Canada |
|---|---|---|
| CCSM4 | 1° x 1.25° | NCAR, University Corporation for Atmospheric Research, U.S.A. |
| CESM1-BGC | 1° x 1.25° | National Science Foundation/DOE, NCAR, U.S.A. |
| CESM1-CAM5 | | |
| CESM1-WACCM | 1.8° x 2.5° | |
| GFDL-ESM2G | 2° x 2.5° | NOAA, Geophysical Fluid Dynamic Laboratory, U.S.A. |
| GFDL-ESM2M | | |
| GISS-E2-H | 2° x 2.5° | NASA, Goddard Institute for Space Studies, U.S.A. |
| GISS-E2-H-CC | | |
| GISS-E2-R | | |
| GISS-E2-R-CC | | |
| HadGEM2-CC | 1.2° x 1.8° | Meteorological Office Hadley Centre, U.K. |
| HadGEM2-ES | | |
| IPSL-CM5A-LR | 1.8° x 3.75° | Laboratoire de Meteorologique Dynamique, Institut Pierre-Simon Laplace, France |
| IPSL-CM5A-MR | 1.25° x 2.5° | |
| IPSL-CM5B-LR | 1.8° x 3.75° | |
| MIROC-ESM | 2.8° x 2.8° | Atmosphere and Ocean Research Institute, National Institute for Environmental Studies, and Japan Agency for Marine-Earth Science and Technology, Japan |
| MIROC-ESM-CHEM | | |
| MPI-ESM-LR | 1.8° x 1.8° | Max Planck Institute for Meteorology, Germany |





| MPI-ESM-MR | | |
|---|---|---|
| NorESM1-M | 1.8° x 1.8° | Norwegian Climate Centre, Norway |
| NorESM1-ME | | |

**Table 3**: Root mean square error between modeled heterotrophic respiration (RH) and pattern predicted RH at the end of the 21[st] century (averaged over 2070-2099) from the RCP 8.5 scenario in Pg C yr$^{-1}$. The patterns used are the $RH_L$ to $RH_G$/$TAS_G$/$NPP_G$.

| Model | $RH_L$-$RH_G$ | $RH_L$-$TAS_G$ | $RH_L$-$NPP_G$ |
|---|---|---|---|
| bcc-csm1-1 | 0.278 | 0.255 | 0.219 |
| bcc-csm1-1-m | 0.319 | 0.260 | 0.259 |
| BNU-ESM | 0.338 | 0.319 | 0.430 |
| CanESM2 | 0.179 | 0.114 | 0.370 |
| CCSM4 | 0.192 | 0.106 | 0.122 |
| CESM1-BGC | 0.188 | 0.112 | 0.129 |
| CESM1-CAM5 | 0.228 | 0.097 | 0.160 |
| CESM1-WACCM | 0.235 | 0.093 | 0.166 |
| GFDL-ESM2G | 0.301 | 0.239 | 0.195 |
| GFDL-ESM2M | 0.430 | 0.209 | 0.593 |
| GISS-E2-H | 0.280 | 0.157 | 0.144 |
| GISS-E2-H-CC | 0.265 | 0.159 | 0.189 |
| GISS-E2-R | 0.135 | 0.127 | 0.091 |
| GISS-E2-R-CC | 0.138 | 0.144 | 0.087 |
| HadGEM2-CC | 0.607 | 0.907 | 0.489 |
| HadGEM2-ES | 0.649 | 0.808 | 0.683 |
| IPSL-CM5A-LR | 0.555 | 0.418 | 0.493 |



| | | | |
|---|---|---|---|
| IPSL-CM5A-MR | 0.457 | 0.326 | 0.435 |
| IPSL-CM5B-LR | 0.239 | 0.369 | 0.186 |
| MIROC-ESM | 0.624 | 0.650 | 0.511 |
| MIROC-ESM-CHEM | 0.708 | 0.666 | 1.012 |
| MPI-ESM-LR | 0.240 | 0.375 | 0.178 |
| MPI-ESM-MR | 0.377 | 0.234 | 0.421 |
| NorESM1-M | 0.130 | 0.090 | 0.083 |
| NorESM1-ME | 0.226 | 0.113 | 0.208 |
| **Ensemble Average** | 0.220 | 0.188 | 0.236 |

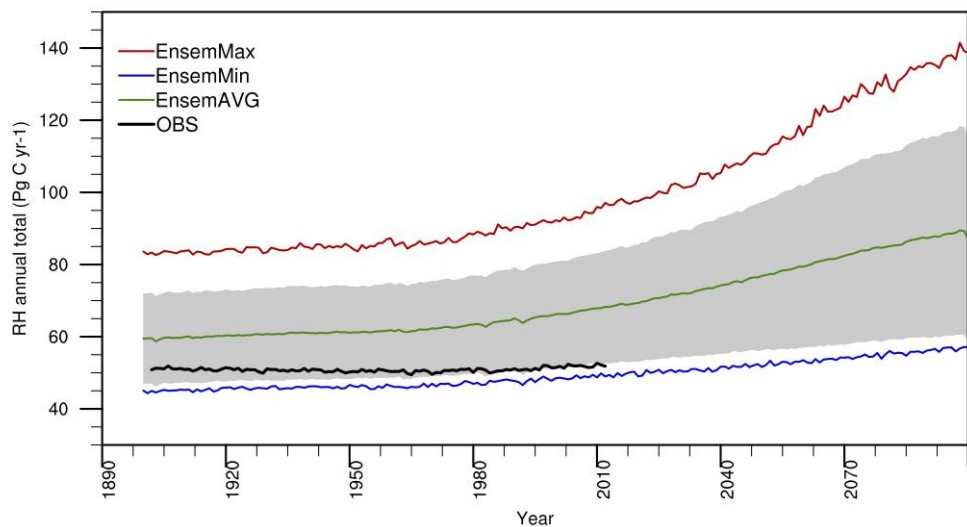

5 **Figure 1:** Annual global heterotrophic respiration (RH) for 25 CMIP5 models, combining the RCP 8.5 and historical outputs. The green line is the ensemble mean; grey shading is ±1 σ of multi-model ensemble mean; red and blue lines show the ensemble maximum and minimum values, respectively. Black line is the 'observed' global annual RH from 1901-2100 from Hashimoto et al. (2015).





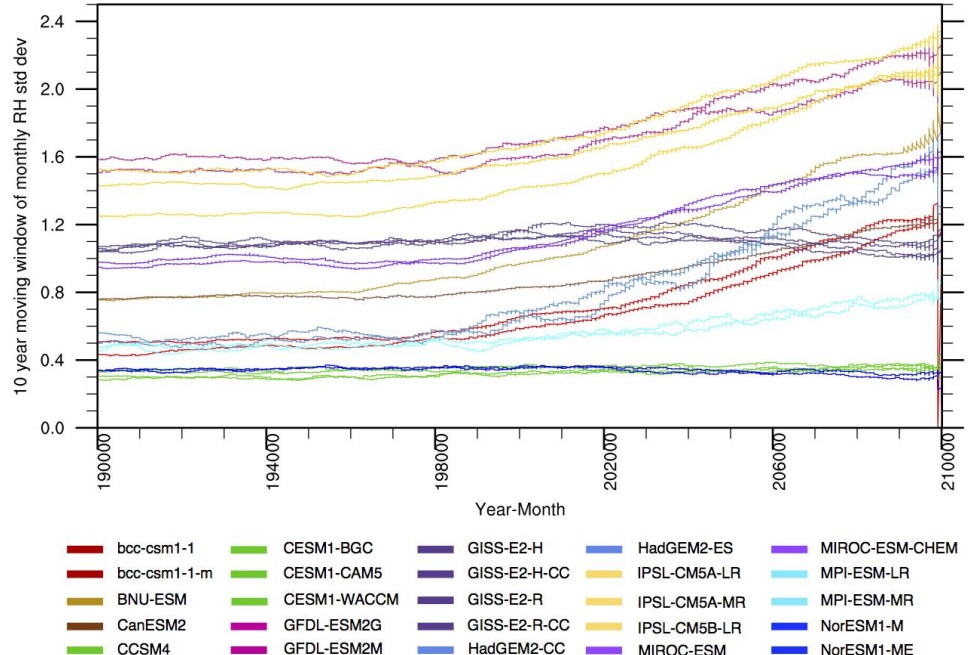

**Figure 2**: Global heterotrophic respiration (RH) inter-annual standard deviation from each model in months since January 1900. A 10-year (120 month) moving window was applied to monthly values from 1900 to 2100 from historical and RCP 8.5 scenarios. Line colors correspond to models from same modeling center.



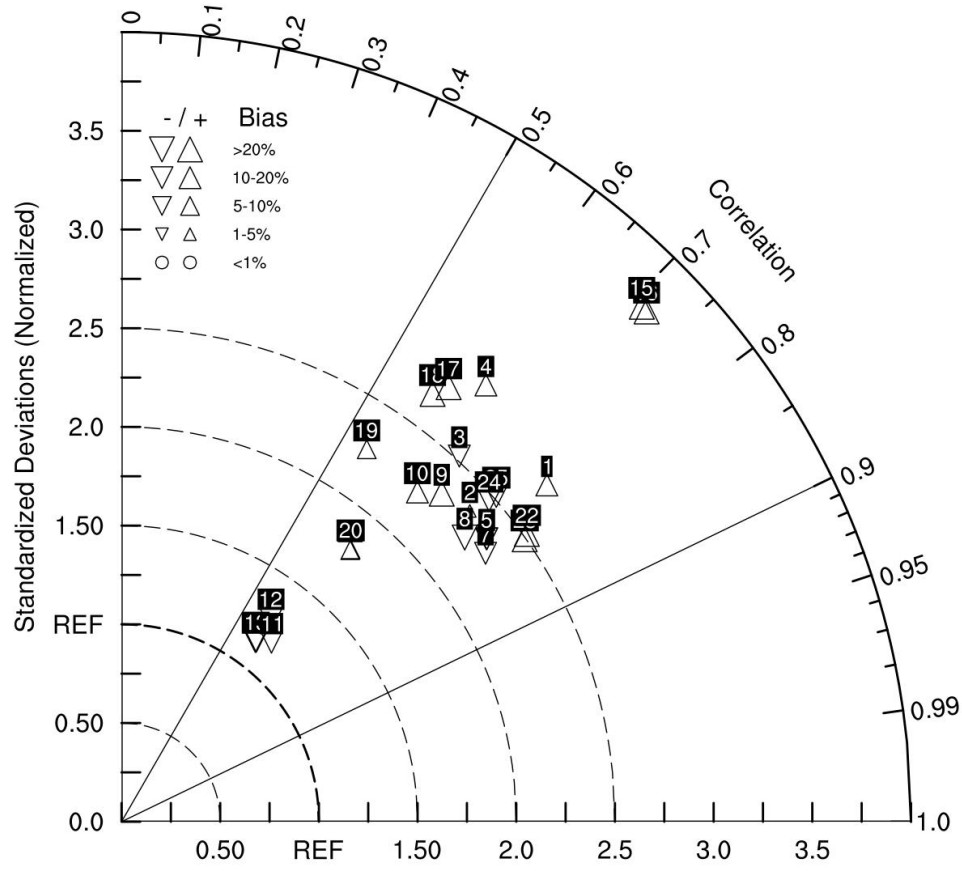

**Figure 3**: Taylor plot of model heterotrophic respiration (RH) bias, normalized root-mean-square (RMS) differences and global spatial correlation as compared to observed RH (Hashimoto et al. 2015). Model data and observations were temporally averaged over the 1950-1999 period; a weighted area sum (80°N-60°S) was done for models and observations.





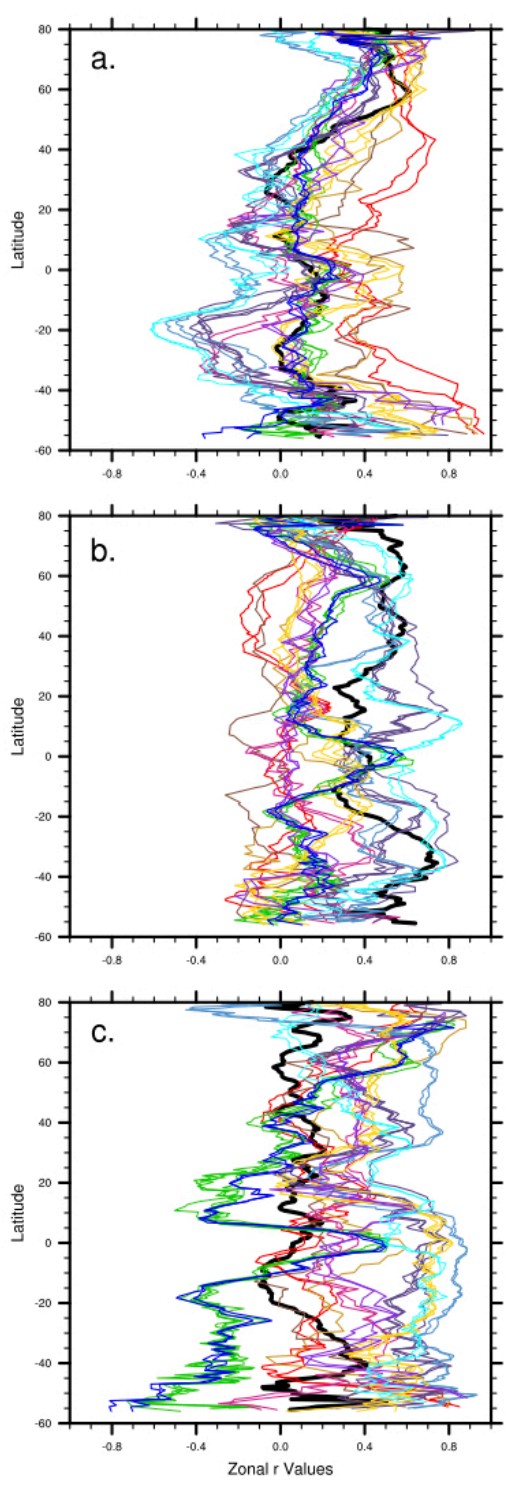




**Figure 4.** Pearson correlation coefficient (*r*) between historical annual heterotrophic
respiration (RH) and surface temperature (TAS, a), precipitation flux (PR, b), and net
primary production (NPP, c) by latitude. Correlation values were calculated at each grid
cell from annual RH and TAS. Black line is the observed RH and TAS (a, 1948-2010),
5    PR (b, 1979-2010), and NPP (c, 2000-2013) *r* values. Line colors correspond to models
from same modeling center and are *r* values for the years 1900-2005. Line colors are
consistent with Figure 2.

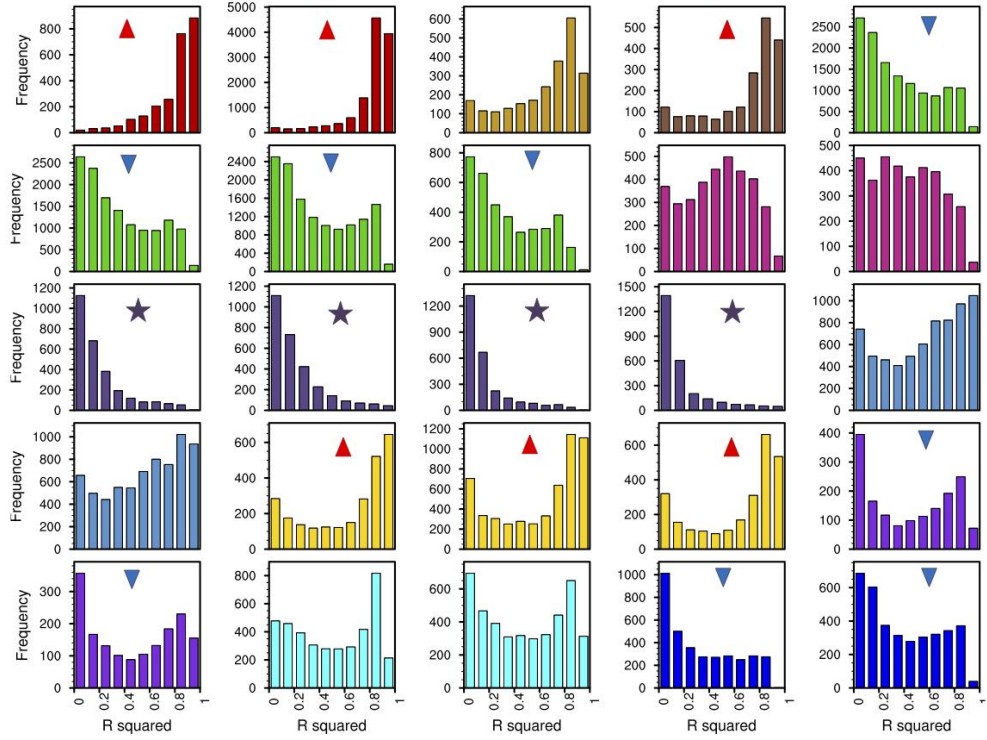

**Figure 5**: Histograms of grid cell $R^2$ values for the relationship between local and global
heterotrophic respiration for RCP 8.5 scenario (2006-2099). Models were grouped by
average $R^2$ into strong relationship (average $R^2 \geq 0.75$, red triangles), weak relationship
(average $R^2 \leq 0.25$, blue triangles), and the GISS-E2 models (dark purple stars). Color of
15    histograms correspond to models from same modeling center. Only values between 80°N
and 60°S are displayed.





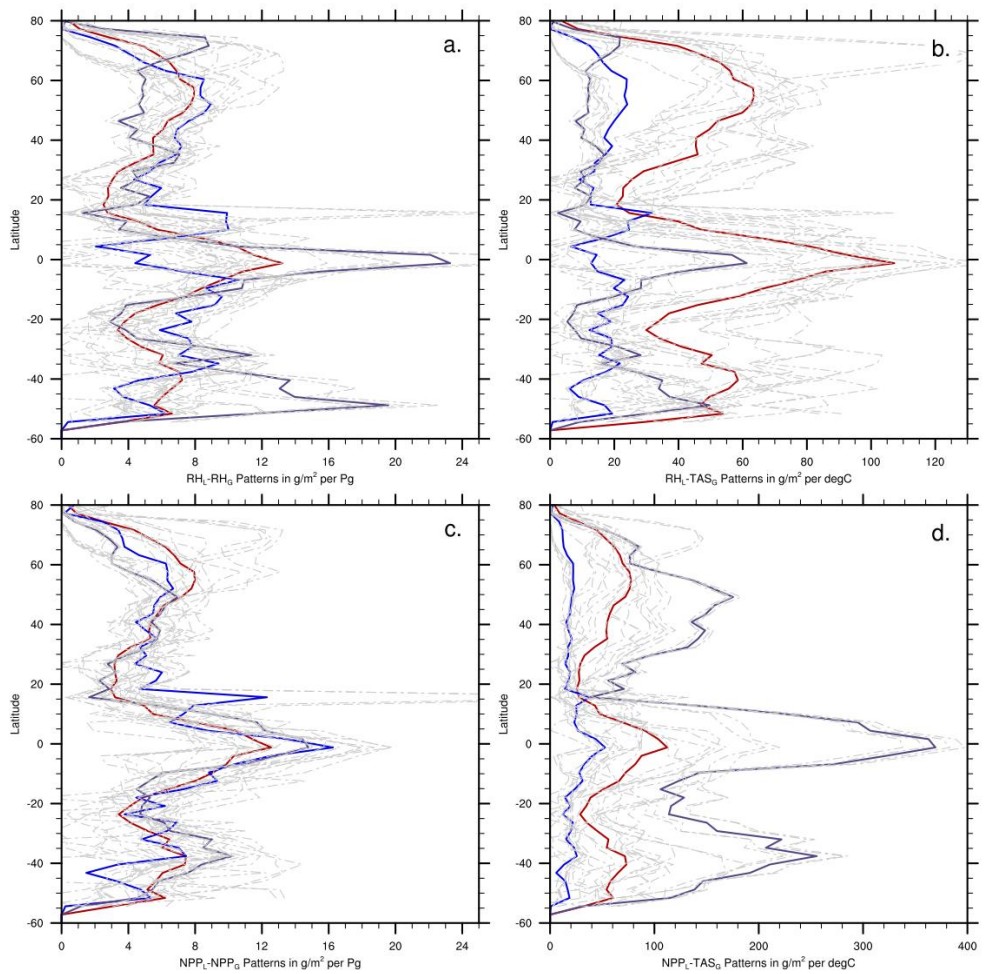

**Figure 6**. Patterns of local-global heterotrophic respiration ($RH_L$-$RH_G$; a), $RH_L$ and global temperature ($TAS_G$; b), local-global net primary productivity ($NPP_L$-$NPP_G$; c), and $NPP_L$ - $TAS_G$ (d) for RCP 8.5 scenario (2006-2099). Models were grouped by average $R^2$ into strong relationship (average $R^2 \geq 0.75$, red line), weak relationship (average $R^2 \leq 0.25$, blue line), and the GISS-E2 models (dark purple line). Gray dashed lines are the patterns from each ensemble member. Only values between 80°N and 60°S are displayed.





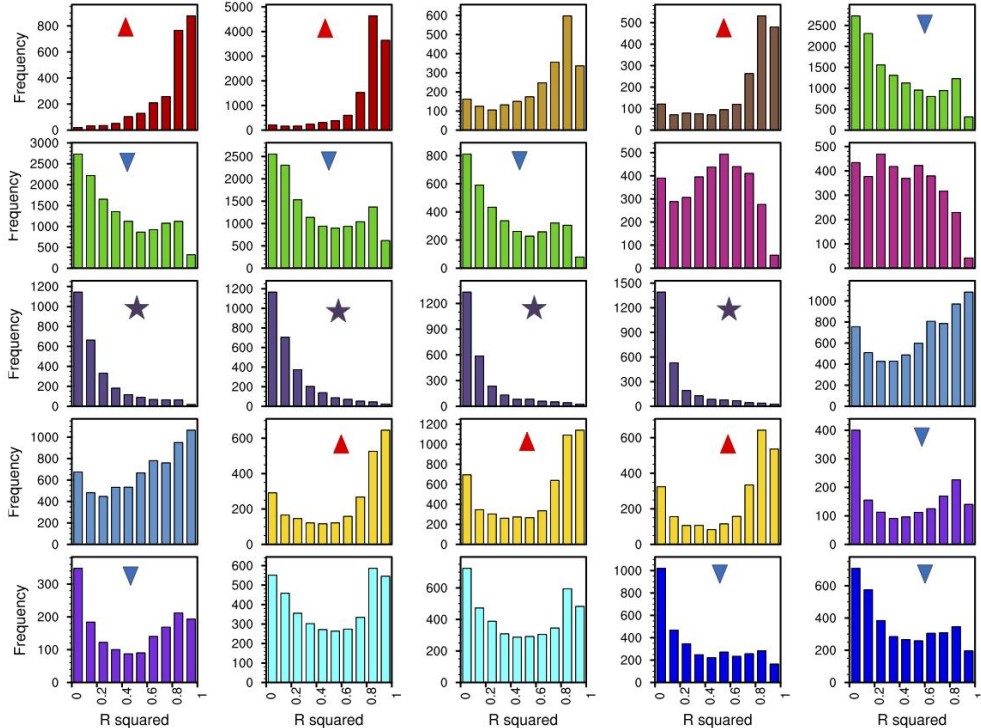

**Figure 7.** Histograms of grid cell $R^2$ values for the relationship between local heterotrophic respiration and global surface temperature ($RH_L$-$TAS_G$) for RCP 8.5 scenario (2006-2099). Histogram colors and groupings are the same as in Figure 5.

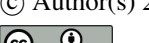



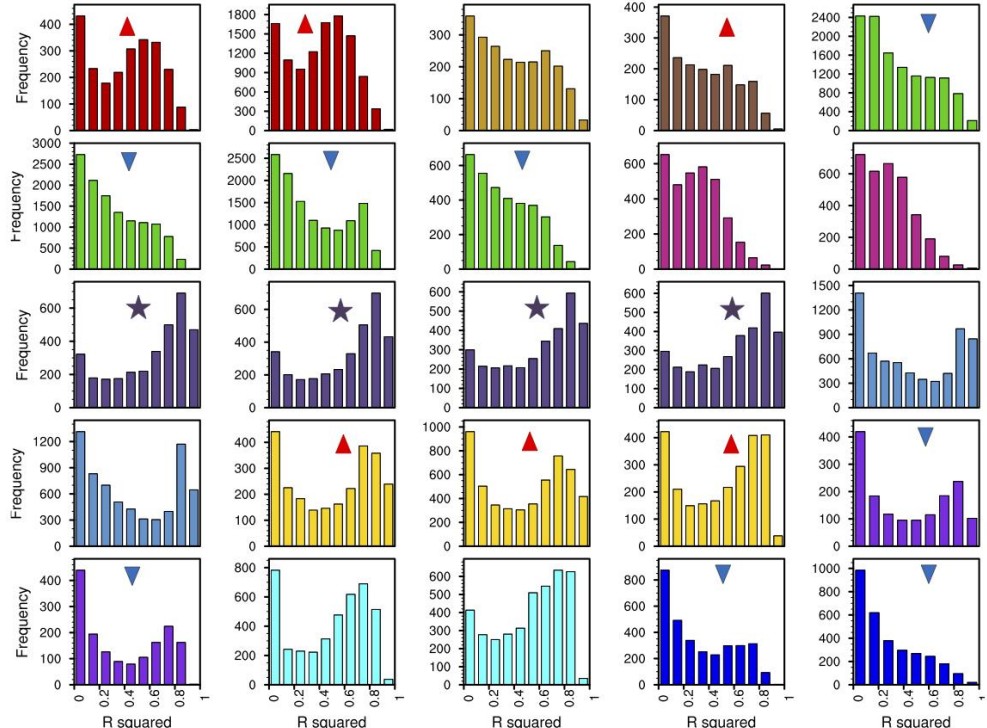

**Figure 8**: Histograms of grid cell $R^2$ values for the relationship between local and global net primary production ($NPP_L$-$NPP_G$) for RCP 8.5 scenario (2006-2099). Histogram colors and groupings are the same as Figure 5.



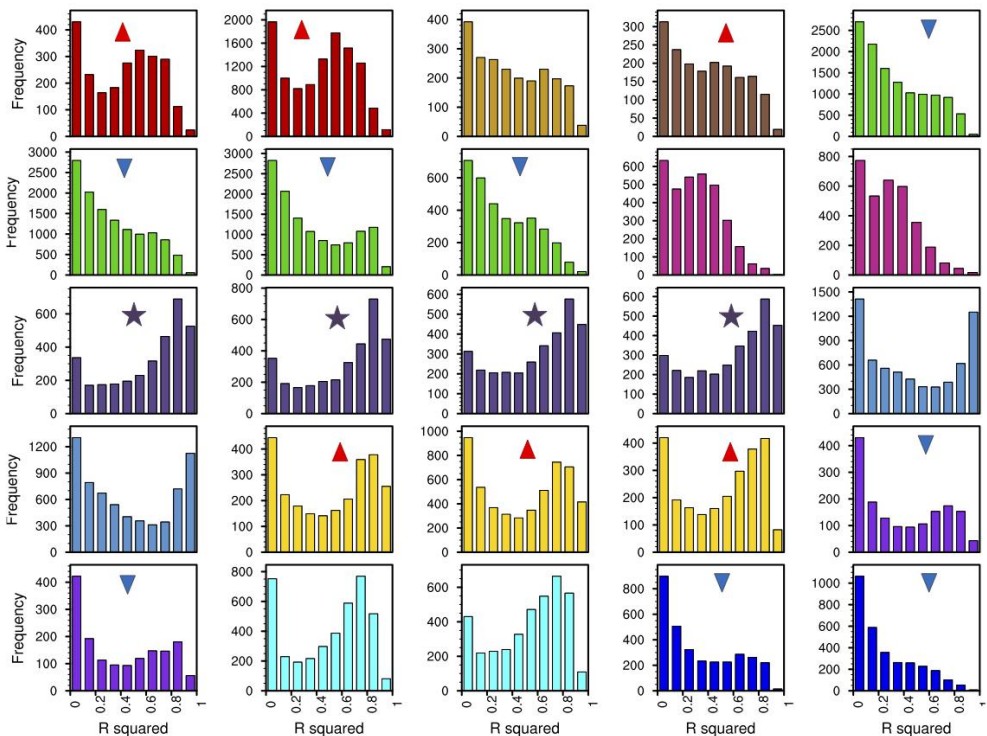

**Figure 9**: Histograms of grid cell $R^2$ values for the relationship between local net primary production and global surface temperature ($NPP_L$-$TAS_G$) for RCP 8.5 scenario (2006-2099). Histogram colors and groupings are the same as Figure 5.



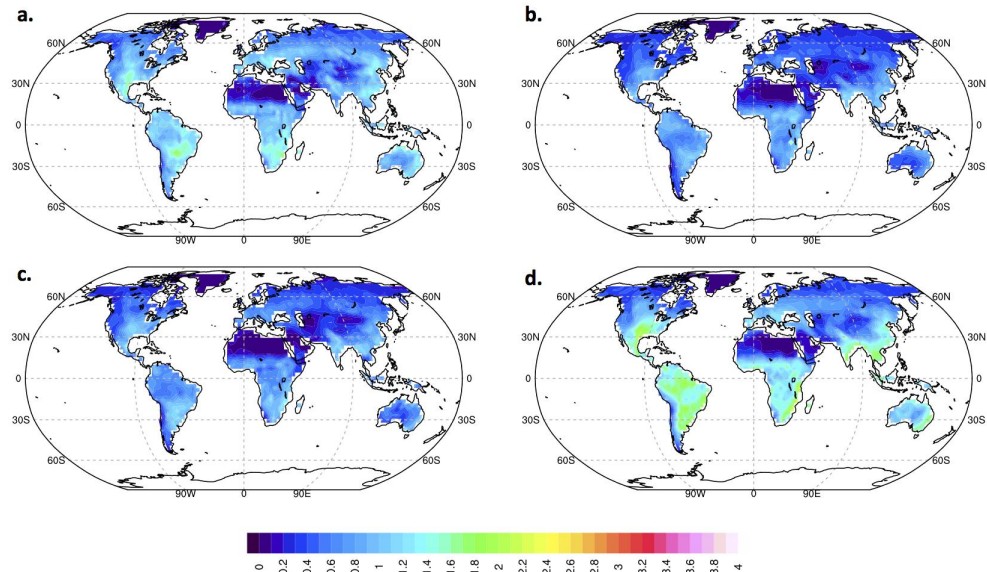

**Figure 10:** Multi-model ensemble average of the standard error of the estimated pattern
for local-global heterotrophic respiration ($RH_L$-$RH_G$ in g/m$^2$ per Pg; a), local RH and
global surface temperature ($RH_L$-$TAS_G$ in g/m$^2$ per °C; b), local RH and global net
primary production ($RH_L$-$NPP_G$ in g/m$^2$ per Pg; c), and local-global net primary
production ($NPP_L$-$NPP_G$ in g/m$^2$ per Pg; d) over 21$^{st}$ century from the RCP 8.5 scenario.