# Peer review of "Causes of uncertainty in observed and projected heterotrophic respiration from Earth System Models"

_Biogeosciences, 2017_

## Referee Comment (RC1) · Anonymous Referee #1 · 9 Nov 2017

This study analyzed heterotrophic respiration (RH) from Earth System Models (ESM) using "pattern scaling", and sought the causes of variations in ESMs' RH output. The terrestrial carbon cycle from ESMs should be tested and constrained by multiple data or multiple carbon flows and stocks; therefore, I agree to the importance of RH in the terrestrial carbon cycle and its simulation. In addition, the application of pattern scaling to RH is new, as far as I know.

Overall, what the authors did is simple: they analyzed RH outputs from ESM using pattern scaling, and discuss the similarity/differences among ESMs. However, even after several readings of this manuscript, I could not follow the details of this manuscript, and could not capture "the causes of uncertainty in observed and projected RH". Rather, the present manuscript appear to simply demonstrate the uncertainty/variations in RH

from ESMs. I hope Results and Discussion should be reorganized. In particular, Discussion is too long (please shorten Discussion and more focus on what you really want to say), and the figures are not of publishable quality.

General comments

Pattern scaling is not widely known outside of climate research field. The authors have to describe the methodology, advantage/disadvantage, and what the results mean more.

Quality of the Figures

The figures are not clear and are not easy to follow. One reason is that the color is the only identifier of each model. The color, however, is not clear. The authors often pointed a specific model name, but for me very difficult to follow. In addition, suddenly, the model number was used.

Figure 5, 7, 8, 9

I am not sure if these figures are necessary. These may be moved to supplement, and/or you can make tables to show the results more clearly.

One serious concern, which I hope is simply my misunderstanding, is that the calculated global RH values seems higher than those I calculated earlier. Did you multiply both areacella (cell area) and sftlf (land surface fraction)? Please check it.

There are several studies about RH in CMIP5 (for example, by Exbrayat et al.), maybe relating your study with those previous studies would be important.

Specific comments:

Page 4, line 42: "even decrease"

Do you have any idea why?

Page 5, line 22-28

Can you be more quantitative?

Page 5, line 28-31

I cannot understand this attribution. Could you elaborate this more carefully?

Page 8, line 3-21

Please compare your results with those by Todd-Brown et al. 2013 more carefully. For example, the MIROC model did not overestimate NPP, why can you say HadGEM2 and IPSL models performed well in capturing observed RH characteristics based on Table 3 in Todd- Brown et al. 2013?

Page 9, line 1: "the observed relationship between RH and NPP is not strong"

Where did this come from?

Page 9, line 33

Extra parenthesis.

Figure 2:

The colors are indistinguishable. The x axis should be improved (190000?)

Figure 3:

The number means what (probably model number)?

Figure 4:

The colors are indistinguishable. The legend should be placed as well.

Figure 5, 7, 8, 9

Why don't you put each model name on each panel?

---

## Referee Comment (RC2) · Anonymous Referee #2 · 14 Jan 2018

General comments:

This manuscript uses a pattern scaling approach to compare CMIP5 models and observational data related to heterotrophic respiration (RH). In terms of absolute RH, on average the models substantially overestimate global RH; the models also predict a substantial increase in RH over the next century. Some but not all models predict an increase in temporal variability of RH as well. Models show spatial biases and are not particularly well correlated with the dataset. RH correlations with temperature, precipitation, and NPP vary zonally in the models, but not in the same way as the observations. Furthermore, models vary dramatically in their local RH sensitivity to global changes in RH, NPP, and climate variables.

I think it is a useful and important exercise to compare ESM outputs of RH. This analysis is novel and complements previous work on soil carbon stocks and NPP. That said, the paper could benefit from an improved explanation of its goals, expectations, and approaches.

Goals: At the end of the intro, the goal is to determine if pattern scaling can be used to evaluate models. Of course it can. But more importantly, can it yield new insight into the underlying biases or problems with the models? Hopefully so, and the intro could do a better job at identifying the issues that are most likely to be revealed by the technique.

Expectations: It was very difficult to interpret the myriad results in the paper without a better indication of how to use the results for model diagnosis. This problem could be addressed by including some pseudo-hypotheses about expected patterns in the model comparisons. For instance, if the processes in a model are overly sensitive to temperature, then what pattern in Fig. 4a might be expected? It was really hard to know what a reader should be looking for when viewing the results, especially with so many different models.

Approaches: I was not previously familiar with the pattern scaling approach, and although it looks powerful, I had a hard time understanding it from the manuscript text. The methods section on pattern scaling could use some elaboration. In particular, I struggled to understand how a single value could be used as the X with multiple Y values in a regression analysis. I was also unclear about the temporal change component. Was the regression relating the change in Y with the change in X over some time interval? The equation presented in this section needs to be explained more clearly or in more detail (or both).

Specific comments:

I wonder if the authors should consider the implications of non-independence among the models. Previously it has been found that ESMs with the same underlying biogeochemical model have very similar predictions of soil carbon spatial distributions and

are not independent. Clearly not all 25 model variants in this paper are independent as some of them generate essentially identical outputs. Is it necessary to show all 25 models? Can they be grouped or aggregated in some sensible way?

Finally, the manuscript seemed to be missing an overall conclusion about the models and recommendations for future model development. There are a lot of discrepancies with the data and across models identified in the paper. Where should the ESM community be moving with respect to improving predictions of RH? It seemed like some models, like GISS, were outliers, but are there other areas that need attention?

Editorial comments:

Abstract fails to give a set of general conclusions specific to this study analysis.

1:41- Not clear what "This" refers to; here an throughout, specify directly.

2:34- I suggest a different formulation of the objective. Almost anything "can" be done. I suspect you were interested in specific aspects of the changes and uncertainty. Can this objective be more informative?

2:46- fix reference formatting

5:39- "does not"

6:32- missing "is"

8:24- missing a word in here somewhere

9:33- fix reference formatting

---

## Referee Comment (RC3) · Anonymous Referee #3 · 17 Jan 2018

**Overview**

The manuscript provides an assessment heterotrophic decomposition simulated by CMIP5 models to temperature and soil moisture (using precipitation as proxy) and how these sensitivities vary in space. Simulation of heterotrophic respiration remains a highly uncertain process in many models and thus any analysis which aims to diagnose the strengths, weaknesses and identifies strategies for improvement are valuable. However, in this case I find the manuscript misses many key areas of existing research in both the introduction and discussion. The writing clarity needs to be improved throughout the manuscript to make the reading as easy as possible. Unfortunately these issues leave me unclear as to what novel information is brought to the fore by this analysis. I hope the authors are able to clarify their message and highlight their

novel finding.

General comments

Writing style:

The authors frequently use overly complex and long sentences with many comma. This makes the manuscript more difficult to read as it frequently obscures the key point of the sentence / paragraph. Below are some examples:

P1 L15: "There is little understanding, however, of the causes of this variability and its consequences for future model development and scenarios evaluation, and examining the relationships between RH and key climate variables may help to understand where and why models are divergent" Would be clearer if broken down e.g. "There is little understanding of the causes of this variability and its consequences for future model development and scenario evaluation. <However,>Examining the relationships between RH and climate variables may help to understand where and why models are divergent."

P1 L27: "The RH-TAS relationship explored here, and more pattern scaling methods mode generally, can be used to efficiently explore uncertainty and projected changes in RH under a wide range of future emission scenarios, and understand how models' structural and parametric choices produce divergent results." Would also be clearer if broken down e.g. "The RH-TAS relationship explored here can be used to efficiently explore uncertainty and projected changes in RH under a wide range of future emission scenarios. Such information is essential to understand how models' structural and parametric choices produce divergent results."

P1 L30 & P8 L25: You should not begin a sentence and definitely not a paragraph with "Because".

Title:

I think that the title should be changes as it is misleading. The manuscript does not

assess the causes of uncertainty in observations as far as I can see. It might be more useful to mention both the CMIP5 models and temperature / precipitation.

Abstract:

The abstract needs greater clarity. It is not clear what if any recommendation are made as to which processes may be missing in the CMIP5 models. What is the pathway to improvement? The authors state that their approach can be used to diagnose causes for divergent results but it is not clear why they do not present any causes for the divergent results found here in the abstract.

P1 L13: It would be good it include a quantification of RH to put into context.

P2 L22: "...RH dataset." This is a little misleading. As the authors point out this is a observation-driven analysis. P1 L25-27: "The relationship between observed RH and precipitation (PR) relationship is strong and positive (r > 0.5, P < 0.005), but few models consistently show this sensitivity of RH and PR." Are the models which do not show a correlation those which do not include a soil moisture response to RH? How many model fail to show the observed behaviour?

Introduction:

The writing style needs addressing. There appears to be some large areas of the existing literature missing from the introduction which is needed to support their analysis. The authors also miss existing literature attempting to diagnose the decomposition processes in CMIP5 models. This information is needed to more clearly define the novelty of the authors work.

The second paragraph of the introduction states most models simulate increasing RH and that existing RH process representation is simple (first order kinetics) compared to many others ecosystem processes (I assume e.g. photosynthesis?). Then moving on to compare observation driven estimate of RH with NPP estimates. I think this is too many concepts in one paragraph without adequately describing any of them. Para-

graph two should deal with a description of exiting RH model structures. Highlighting known issues with first order kinetic, e.g. lack of a microbial pool and difficulties in responding to changes in litter quality versus quantity (e.g. Wieder et al., 2013, Xenakis & Williams 2014). The importance of soil moisture (Exbrayat et al., 2013ab, Exbrayat et al., 2014) or nutrient cycles (Manzoni & Porporato 2009; Exbrayat et al., 2013a).

P1 L44-45 "While both temperature and precipitation have a positive effect on the global terrestrial carbon flux" Are you still talking about respiration? Net ecosystem exchange, Net biome exchange?

Methods:

Linking back to the decomposition review from the introduction details of which temperature and soil moisture response functions used in the CMIP5 models seems appropriate to me.

P3 L1-6: Does the observation-driven estimates come with an uncertainty analysis?

P3 L21: "...we only used the first realisation..." Would it not be more appropriate to use the mean across ensembles?

Results:

P5 L14-20: It is not immediately clear whether you are talking about correlations in space or time. Also please be clear throughout the manuscript that precipitation is a proxy for soil moisture availability. Therefore you should not be talking about both soil moisture and rainfall being limiting. Soil moisture / plant available water is limited.

P5 L28-30: "This is likely due to less land (and this higher variability is model averages)..." Is it not equally or more likely that greater divergence between models occurs because the models are trained and developed using observations which are bias to the temperate northern hemisphere?

P6 L13-21: Is there no observation equivalent for this analysis?

P7 L1-14: I think both of these paragraphs need to be clearer. It is not always obvious to me whether you are talking about simulated vs observed RH. Would be improved if you make better use of the available figures in this section.

Discussion:

P8 L19-20: This is the first time pools have been mentioned. This should be first mentioned in the introduction.

Moreover, I feel a more thorough discussion of the associated literature is needed.

P8 L25-46: I think both these paragraphs need rephrasing to improve clarity. There is no use of any figures or tables from you manuscript here.

P9 L15-17: Your text appears to be referring to the global average but what about spatial patterns? You present a large number of figures with spatial variation, can you make greater use of these?

P9 L22-30: I think this would be a good area to discuss some of the possible parameterisation / model processes missing within the existing models within the context of the material I suggested should be added to the introduction.

P9 L39: "...temperature response (Q10)..." please provide range for context.

Specific comments

P1 L22 "Compared to observations, ESMs consistency..." -> "Compared to observations ESMs consistently..."

P1 L36 "carbon cycle" -> "carbon (C) cycle"

P2 L31 "...an observation-based data product." -> "...observation-driven analysis."

P2 L46 "(Hashimoto et al., 2015):(Hashimoto et al., 2015)"

P4 L40: "...majority (65 %)..." please state number of models.

P5 L1: "...smallest projected trend..." trend in what? not clear from text.

P6 L24: "...weaker correlated models..."

P7 L21: "...ESMs examined here (Figure X)" ?

P7 L23: delete "On one hand, "

P7 L25: delete "On the other,"

P7 L30: "...soil moisture response functions."

P7 L35: "robustly" I'm not convinced you can say this. "Consistently" would be a more appropriate word

P8 L1: delete "Interestingly,"

P8 L9: "...across empirical datasets." Such as?

P8 L25: "Because..." you should not begin a sentence with "because", let alone a paragraph or subsection. Please rephrase.

References not included in original manuscript Exbrayat J-F, Pitman AJ, Zhang Q, Abramowitz G, Wang Y-P (2013a) Examining soil carbon uncertainty in a global model: response of microbial decomposition to temperature, moisture and nutrient limitation. Biogeosciences 10:7095-7108. doi: 10.5194/bg-10-7095-2013

Exbrayat J-F, Pitman AJ, Abramowitz G, Wang Y-P (2013b), Sensitivity of net ecosystem exchange and heterotrophic respiration to parameterization uncertainty. Journal of Geophysical Research: Atmospheres 118:1640-1651. doi: 10.1029/2012JD018122

Exbrayat J-F, Pitman AJ, Abramowitz G (2014) Response of microbial decomposition to spin-up explains CMIP5 soil carbon range until 2100. Geoscientific Model Development 7:2683-2692. doi: 10.5194/gmd-7-2683-2014

Xenakis, G., and Williams, M. (2014) Comparing microbial and chemical approaches for modelling soil organic carbon decomposition using the DecoChem v1.0 and Deco-

Bio v1.0 models, Geoscientific Model Development , 7, 1519-1533, doi:10.5194/gmd-7-1519-2014

---

## Referee Comment (RC4) · Anonymous Referee #4 · 19 Jan 2018

General Comments

This manuscript used a global gridded heterotrophic respiration (RH) obtained from assimilating observed soil respiration into a statistical model to benchmark 25 CMPI5 Earth System Models (ESMs) in simulating RH, globally. The overall RH trends simulated by CMIP5 models are displayed, and possible reasons for the discrepancies between "observations" and modeling results are discussed. The topic of the manuscript is timely, as the RH simulation is not well represented in ESMs. However, there are several parts that are hard to understand. I recommend the authors to clarify the issues included in my comments.

Recommendation

Major revision.

Major comments

1. P3, L20: The authors used “all available climate models”, and then generated “an ensemble of 25 ESMs”. Then, they mentioned “the first ensemble member from each model”. How many ensemble member of each model? If each model (e.g., CESM1-BGC) has several ensemble members, why the authors did not use the mean of the ensemble members? I think this part needs to be clarified.

2. I did not totally understand the “pattern scaling” method even though the equation is shown. Why particularly this method is used in the manuscript? What is the advantage of this method? In addition, Table 3 listed single numbers of, for example, RHL-RHG, but Figure 6 displayed the meridional variations of the similar relationships. Was Tables 3 the global mean of Figure 6? If so, what is the physical meaning of calculating the global mean of the RHL-RHG relationship? A better explanation of this method and the related results are needed.

3. P5, L16, can the authors show the proofs, saying papers, discussing “the dominant control on RH is temperature in these cold biomes”?

4. P5, L40, I suggest the authors to give examples of the models either overestimated or underestimated the r values.

5. P9, L26, what is the theoretical basis of the “RH-NPP relationship” in different ecosystems?

6. P9, L37, can the authors specify the climate factors here? In other words, besides temperature, what are the factors regulating carbon decomposition rates in a soil water limited environment? In P5, L16, the authors mentioned that “the dominant control on RH is temperature in these cold biomes”. Looking at these two sentences together, does it mean that 1) in reality temperature is the main factor controlling RH in cold regions; 2) the sensitivity of soil carbon decomposition, closely related to RH, to temperature in cold regions is limited by soil moisture in ESMs? If so, can the authors explain the reasons for the difference between reality and models?

Minor comments

1. RH is used as the acronym of "heterotrophic respiration". In my view, it should be HR, and it is easily to think RH as "relative humidity", especially for a paper related to different climate factors. It is fine if most of the papers define "heterotrophic respiration" as RH. Otherwise, please correct it.

2. P5, L14, TAS was named before, and does not need to be re-named. In addition, surface air temperature and surface temperature (not TAS) are two temperature definitions. The authors need to give a clear description here.

3. It is not necessary using PR as the acronym name of precipitation. Also, the authors used PR and precipitation randomly. If an acronym name is defined, it can be used afterward.

4. P5, L21, should it be "few" or "A few"? The authors used a colon here, and it looks to me that CCSM/CESM, to some extend, can capture the patterns.

---

## Author Comment (AC1) · 23 Feb 2018

Please see responses in supplement attached.

Please also note the supplement to this comment:
https://www.biogeosciences-discuss.net/bg-2017-405/bg-2017-405-AC1-supplement.pdf
* * *

---

## Author Comment (AC2) · 23 Feb 2018

**Reviewer 2**
* * *
*General comments:*
*This manuscript uses a pattern scaling approach to compare CMIP5 models and observational data related to heterotrophic respiration (RH). In terms of absolute RH, on average the models substantially overestimate global RH; the models also predict a substantial increase in RH over the next century. Some but not all models predict an increase in temporal variability of RH as well. Models show spatial biases and are not particularly well correlated with the dataset. RH correlations with temperature, precipitation, and NPP vary zonally in the models, but not in the same way as the observations. Furthermore, models vary dramatically in their local RH sensitivity to global changes in RH, NPP, and climate variables.*

*I think it is a useful and important exercise to compare ESM outputs of RH. This analysis is novel and complements previous work on soil carbon stocks and NPP. That said, the paper could benefit from an improved explanation of its goals, expectations, and approaches.*

Thanks for the careful reading and useful feedback. We agree that this manuscript needs significant revisions in many areas, but are hopeful that doing so will greatly improve its clarity, methodological rigor, and ultimately impact.
* * *
*Goals: At the end of the intro, the goal is to determine if pattern scaling can be used to evaluate models. Of course it can. But more importantly, can it yield new insight into the underlying biases or problems with the models? Hopefully so, and the intro could do a better job at identifying the issues that are most likely to be revealed by the technique.*

This is a good point. In our revision we will provide more background on pattern scaling, as well as the state of CMIP5 models' carbon cycle performance more generally (Anav et al., 2013; Luo et al., 2016), and using aspects of model behavior to draw inferences about climate- and carbon-cycle response to anthropogenic forcing (e.g. Gillett et al., 2013). In addition, we think that a better discussion of how RH pattern scaling can be treated as a type of emergent constraint (Hoffman et al., 2014; Luo et al., 2015) would be useful.
* * *
*Expectations: It was very difficult to interpret the myriad results in the paper without a better indication of how to use the results for model diagnosis. This problem could be addressed by including some pseudo-hypotheses about expected patterns in the model comparisons. For instance, if the processes in a model are overly sensitive to temperature, then what pattern in Fig. 4a might be expected? It was really hard to know what a reader should be looking for when viewing the results, especially with so many different models.*

This is an excellent suggestion–thank you. In our revision we will clearly lay out specific hypotheses or at least expectations: how, based on our best understanding of the carbon cycle and scaling issues (Bond-Lamberty et al., 2016; Jung et al., 2017; Phillips et al., 2017), models might be expected to behave across latitudinal and global scales.
* * *
*Approaches: I was not previously familiar with the pattern scaling approach, and although it looks powerful, I had a hard time understanding it from the manuscript text. The methods section on pattern scaling could use some elaboration. In particular, I struggled to understand how a single value could be used as the X with multiple Y values in a regression analysis. I was also unclear about the temporal change component. Was the regression relating the change in Y with the change in X over some time interval? The equation presented in this section needs to be explained more clearly or in more detail (or both).*

In our revision we will provide more background on pattern scaling: its strengths and limitations (Tebaldi and Arblaster, 2014), accuracy (Herger et al., 2015; Mitchell, 2003), and perhaps novel statistical approaches to overcome some its limitations (Link et al., 2018). All equation(s) will be more carefully explained.
* * *
*Specific comments:*
*I wonder if the authors should consider the implications of non-independence among the models. Previously it has been found that ESMs with the same underlying biogeochemical model have very similar predictions of soil carbon spatial distributions and are not independent. Clearly not all 25 model variants in this paper are independent as some of them generate essentially identical outputs. Is it necessary to show all 25 models? Can they be grouped or aggregated in some sensible way?*

We agree, and attempted to do this already (to some degree) in Figure 5. Yes, this is an excellent suggestion and will tie in nicely with the repeated reviewer requests for more clarity in figures and results–we will do much more grouping by 'model center' or land model used (e.g. different variants of CESM, etc).
* * *
*Finally, the manuscript seemed to be missing an overall conclusion about the models and recommendations for future model development. There are a lot of discrepancies with the data and across models identified in the paper. Where should the ESM community be moving with respect to improving predictions of RH? It seemed like some models, like GISS, were outliers, but are there other areas that need attention?*
*Editorial comments:*
*Abstract fails to give a set of general conclusions specific to this study analysis.*

We agree, and will place more emphasis on synthesizing suggestions for future directions and steps to improve RH modeling.
* * *
*1:41- Not clear what "This" refers to; here and throughout, specify directly.*

Thanks; we will do so.
* * *
*2:34- I suggest a different formulation of the objective. Almost anything "can" be done. I suspect you were interested in specific aspects of the changes and uncertainty. Can this objective be more informative?*

This links well with the suggestion above to have specific hypotheses, or at least expectations. We feel that this will help better define a tight objective, which we agree is currently missing.
* * *
*2:46- fix reference formatting*
*5:39- "does not"*
*6:32- missing "is"*
*8:24- missing a word in here somewhere*
*9:33- fix reference formatting*

We will fix these issues.

**References cited in response**

Anav, A., Friedlingstein, P., Kidston, M., Bopp, L., Ciais, P., Cox, P. M., Jones, C. D., Jung, M., Myneni, R. B. and Zhu, Z.: Evaluating the land and ocean components of the global carbon cycle in the CMIP5 earth system models, J. Clim., 26, 6801–6843, 2013.

Bond-Lamberty, B. and Thomson, A. M.: Temperature-associated increases in the global soil respiration record, Nature, 464(7288), 579–582, 2010.

Bond-Lamberty, B., Epron, D., Harden, J. W., Harmon, M. E., Hoffman, F. M., Kumar, J., McGuire, A. D. and Vargas, R.: Estimating heterotrophic respiration at large scales: challenges, approaches, and next steps, Ecosphere, 7(6), d01380, 2016.

Davidson, E. A. and Janssens, I. A.: Temperature sensitivity of soil carbon decomposition and feedbacks to climate change, Nature, 440, 165–173, 2006.

Exbrayat, J.-F., Pitman, A. J., Zhang, Q., Abramowitz, G. and Wang, Y.-P.: Examining soil carbon uncertainty in a global model: response of microbial decomposition to temperature, moisture and nutrient limitation, Biogeosciences, 10, 7095–7108, 2013a.

Exbrayat, J.-F., Pitman, A. J., Abramowitz, G. and Wang, Y.-P.: Sensitivity of net ecosystem exchange and heterotrophic respiration to parameterization uncertainty, J. Geophys. Res., doi:10.1029/2012JD018122, 2013b.

Falloon, P. D., Jones, C. D., Ades, M. and Paul, K. I.: Direct soil moisture controls of future global soil carbon changes: An important source of uncertainty, Global Biochemical Cycles, 25, GB3010, 2011.

Gillett, N. P., Arora, V. K., Matthews, D. and Allen, M. R.: Constraining the Ratio of Global Warming to Cumulative CO2 Emissions Using CMIP5 Simulations, J. Clim., 26(18), 6844–6858, 2013.

Guenet, B., Camino-Serrano, M., Ciais, P., Tifafi, M., Maignan, F., Soong, J. L. and Janssens, I. A.: Impact of priming on global soil carbon stocks, Glob. Chang. Biol., doi:10.1111/gcb.14069, 2018.

Hashimoto, S., Carvalhais, N., Ito, A., Migliavacca, M., Nishina, K. and Reichstein, M.: Global spatiotemporal distribution of soil respiration modeled using a global database, Biogeosciences, 12, 4121–4132, 2015.

Herger, N., Sanderson, B. W. and Knutti, R.: Improved pattern scaling approaches for the use in climate impact studies, Geophys. Res. Lett., in press, doi:10.1002/2015GL063569, 2015.

Hoffman, F. M., Randerson, J. T., Arora, V., Bao, Q., Six, K. D., Cadule, P., Ji, D., Jones, C. D., Kawamiya, M., Khatiwala, S., Lindsay, K., Obata, A., Shevliakova, E., Tjiputra, J., Volodin, E. M. and Wu, T.: Causes and implications of persistent atmospheric carbon dioxide biases in Earth System Models, Journal of Geophysical Research-Biogeosciences, in press, doi:10.1002/2013JG002381, 2014.

Hursh, A., Ballantyne, A., Cooper, L., Maneta, M., Kimball, J. and Watts, J.: The sensitivity of

soil respiration to soil temperature, moisture, and carbon supply at the global scale, Glob. Chang. Biol., 23(5), 2090–2103, 2017.

Jung, M., Reichstein, M., Margolis, H. A., Cescatti, A., Richardson, A. D., Arain, M. A., Arneth, A., Bernhofer, C., Bonal, D., Chen, J., Gianelle, D., Gobron, N., Kiely, G., Kutsch, W. L., Lasslop, G., Law, B. E., Lindroth, A., Merbold, L., Montagnani, L., Moors, E. J., Papale, D., Sottocornola, M., Vaccari, F. and Williams, C. A.: Global patterns of land-atmosphere fluxes of carbon dioxide, latent heat, and sensible heat derived from eddy covariance, satellite, and meteorological observations, Journal of Geophysical Research-Biogeosciences, 116, G00J07, 2011.

Jung, M., Reichstein, M., Schwalm, C. R., Huntingford, C., Sitch, S., Ahlström, A., Arneth, A., Camps-Valls, G., Ciais, P., Friedlingstein, P., Gans, F., Ichii, K., Jain, A. K., Kato, E., Papale, D., Poulter, B., Raduly, B., Rödenbeck, C., Tramontana, G., Viovy, N., Wang, Y.-P., Weber, U., Zaehle, S. and Zeng, N.: Compensatory water effects link yearly global land CO2 sink changes to temperature, Nature, 541(7638), 516–520, 2017.

Le Quéré, C., Andrew, R. M., Friedlingstein, P., Sitch, S., Pongratz, J., Manning, A. C., Korsbakken, J. I., Peters, G. P., Canadell, J. G., Jackson, R. B., Boden, T. A., Tans, P. P., Andrews, O. D., Arora, V. K., Bakker, D. C. E., Barbero, L., Becker, M., Betts, R. A., Bopp, L., Chevallier, F., Chini, L. P., Ciais, P., Cosca, C. E., Cross, J., Currie, K., Gasser, T., Harris, I., Hauck, J., Haverd, V., Houghton, R. A., Hunt, C. W., Hurtt, G., Ilyina, T., Jain, A. K., Kato, E., Kautz, M., Keeling, R. F., Klein Goldewijk, K., Körtzinger, A., Landschützer, P., Lefèvre, N., Lenton, A., Lienert, S., Lima, I., Lombardozzi, D., Metzl, N., Millero, F., Monteiro, P. M. S., Munro, D. R., Nabel, J. E. M. S., Nakaoka, S.-I., Nojiri, Y., Padín, X. A., Peregon, A., Pfeil, B., Pierrot, D., Poulter, B., Rehder, G., Reimer, J., Rödenbeck, C., Schwinger, J., Séférian, R., Skjelvan, I., Stocker, B. D., Tian, H., Tilbrook, B., van der Laan-Luijkx, I. T., van der Werf, G. R., van Heuven, S., Viovy, N., Vuichard, N., Walker, A. P., Watson, A. J., Wiltshire, A. J., Zaehle, S. and Zhu, D.: Global Carbon Budget 2017, Earth Syst. Sci. Data Discuss., 1–79, 2017.

Link, R., Bond-Lamberty, B., Hartin, C., Lynch, C. and Kravitz, B.: Computationally efficient emulators for Earth System Models, Geoscientific Model Development, submitted, 2018.

Liu, L., Wang, X., Lajeunesse, M. J., Miao, G., Piao, S., Wan, S., Wu, Y., Wang, Z., Yang, S. and Deng, M.: A cross-biome synthesis of soil respiration and its determinants under simulated precipitation changes, Glob. Chang. Biol., 22(4), 1394–1405, 2016.

Luo, Y., Keenan, T. F. and Smith, M.: Predictability of the terrestrial carbon cycle, Glob. Chang. Biol., 21(5), 1737–1751, 2015.

Luo, Y., Ahlström, A., Allison, S. D., Batjes, N. H., Brovkin, V., Carvalhais, N., Chappell, A., Ciais, P., Davidson, E. A., Finzi, A., Georgiou, K., Guenet, B., Hararuk, O., Harden, J. W., He, Y., Hopkins, F., Jiang, L., Koven, C., Jackson, R. B., Jones, C. D., Lara, M. J., Liang, J., McGuire, A. D., Parton, W., Peng, C., Randerson, J. T., Salazar, A., Sierra, C. A., Smith, M. J., Tian, H., Todd-Brown, K. E. O., Torn, M., van Groenigen, K. J., Wang, Y. P., West, T. O., Wei, Y., Wieder, W. R., Xia, J., Xu, X., Xu, X. and Zhou, T.: Toward more realistic projections of soil carbon dynamics by Earth system models, Global Biogeochem. Cycles, 30(1), 2015GB005239, 2016.

Mahecha, M. D., Reichstein, M., Carvalhais, N., Lasslop, G., Lange, H., Seneviratne, S. I., Vargas, R., Ammann, C., Arain, M. A., Cescatti, A., Janssens, I. A., Migliavacca, M.,

Montagnani, L. and Richardson, A. D.: Global convergence in the temperature sensitivity of respiration at ecosystem level, Science, 329(5993), 838–840, 2010.

McGuire, A. D., Anderson, L. G., Christensen, T. R., Dallimore, S., Guo, L., Hayes, D. J., Heimann, M., Lorenson, T. D., Macdonald, R. W. and Roulet, N. T.: Sensitivity of the carbon cycle in the Arctic to climate change, Ecol. Monogr., 79(4), 523–555, 2009.

Mitchell, T. D.: Pattern Scaling: An Examination of the Accuracy of the Technique for Describing Future Climates, Clim. Change, 60(3), 217–242, 2003.

Moyano, F. E., Manzoni, S. and Chenu, C.: Responses of soil heterotrophic respiration to moisture availability: An exploration of processes and models, Soil Biol. Biochem., 59, 72–85, 2013.

Odum, E. P.: The strategy of ecosystem development, Science, 164, 262–270, 1969.

Phillips, C. L., Bond-Lamberty, B., Desai, A. R., Lavoie, M., Risk, D., Tang, J., Todd-Brown, K. and Vargas, R.: The value of soil respiration measurements for interpreting and modeling terrestrial carbon cycling, Plant Soil, 413(1-2), 1–25, 2017.

Rehfeld, K., Münch, T., Ho, S. L. and Laepple, T.: Global patterns of declining temperature variability from the Last Glacial Maximum to the Holocene, Nature, 554(7692), 356–359, 2018.

Schimel, D., Stephens, B. B. and Fisher, J. B.: Effect of increasing CO2 on the terrestrial carbon cycle, Proceedings of the National Academy of Science, 112(2), 436–441, 2015.

Shao, P., Zeng, X., Moore, D. J. P. and Zeng, X.: Soil microbial respiration from observations and Earth System Models, Environ. Res. Lett., 8(3), 034034, 2013.

Tebaldi, C. and Arblaster, J. M.: Pattern scaling: Its strengths and limitations, and an update on the latest model simulations, Clim. Change, 122(3), 459–471, 2014.

Todd-Brown, K. E. O., Randerson, J. T., Post, W. M., Hoffman, F. M., Tarnocai, C., Schuur, E. A. G. and Allison, S. D.: Causes of variation in soil carbon predictions from CMIP5 Earth system models and comparison with observations, Biogeosciences, 10, 1717–1736, 2013.

Xiao, J., Davis, K. J., Urban, N. M., Keller, K. and Saliendra, N. Z.: Upscaling carbon fluxes from towers to the regional scale: Influence of parameter variability and land cover representation on regional flux estimates, Journal of Geophysical Research-Biogeosciences, 116, GB3027, 2011.

---

## Author Comment (AC3) · 23 Feb 2018

**Reviewer 3**
* * *
*Overview*
*The manuscript provides an assessment heterotrophic decomposition simulated by CMIP5 models to temperature and soil moisture (using precipitation as proxy) and how these sensitivities vary in space. Simulation of heterotrophic respiration remains a highly uncertain process in many models and thus any analysis which aims to diagnose the strengths, weaknesses and identifies strategies for improvement are valuable. However, in this case I find the manuscript misses many key areas of existing research in both the introduction and discussion. The writing clarity needs to be improved throughout the manuscript to make the reading as easy as possible. Unfortunately these issues leave me unclear as to what novel information is brought to the fore by this analysis. I hope the authors are able to clarify their message and highlight their novel finding.*

Thanks for the careful reading and useful feedback. We agree that this manuscript needs significant revisions in many areas, but are hopeful that doing so will greatly improve its clarity, methodological rigor, and ultimately impact.
* * *
*General comments*
*Writing style:*
*The authors frequently use overly complex and long sentences with many comma. This makes the manuscript more difficult to read as it frequently obscures the key point of the sentence / paragraph. Below are some examples...*
*P1 L30 & P8 L25: You should not begin a sentence and definitely not a paragraph with "Because".*

Thank you. We agree that these need simplification and clarification, and in many other spots as well. Awkward grammar and conjunction oddities will be fixed as well in our revision.
* * *
*Title:*
*I think that the title should be changes as it is misleading. The manuscript does not assess the causes of uncertainty in observations as far as I can see. It might be more useful to mention both the CMIP5 models and temperature / precipitation.*

Thanks for the suggestion. We will look carefully at the title after fully revising the manuscript and assess its suitability, precision, and clarity.
* * *
*Abstract:*

*The abstract needs greater clarity. It is not clear what if any recommendation are made as to which processes may be missing in the CMIP5 models. What is the pathway to improvement?*

We agree, and will place more emphasis on clarity and synthesizing suggestions for future directions and steps to improve RH modeling.
* * *
*The authors state that their approach can be used to diagnose causes for divergent results but it is not clear why they do not present any causes for the divergent results found here in the abstract.*

This is an good point. In our revision we will clearly lay out specific hypotheses or at least expectations: how, based on our best understanding of the carbon cycle and scaling issues (Bond-Lamberty et al., 2016; Jung et al., 2017; Phillips et al., 2017), models might be expected to behave across latitudinal and global scales, and why they might diverge based on parametric or structural differences.
* * *
*P1 L13: It would be good it include a quantification of RH to put into context.*

We agree, and will do so.
* * *
*P2 L22: "...RH dataset." This is a little misleading. As the authors point out this is a observation-driven analysis.*

We believe the reviewer means line 42, not 22? Good point; we will clarify this, drawing a distinction between true observations and upscale datasets *derived* from observations.
* * *
*P1 L25-27: "The relationship between observed RH and precipitation (PR) relationship is strong and positive (r > 0.5, P < 0.005), but few models consistently show this sensitivity of RH and PR." Are the models which do not show a correlation those which do not include a soil moisture response to RH? How many model fail to show the observed behaviour?*

This is an interesting suggestion, and one that we will check and quantify, referencing recent work in this area (Falloon et al., 2011; Liu et al., 2016; Moyano et al., 2013).
* * *
*Introduction:*
*The writing style needs addressing. There appears to be some large areas of the existing literature missing from the introduction which is needed to support their analysis. The authors also miss existing literature attempting to diagnose the decomposition processes in CMIP5*

*models. This information is needed to more clearly define the novelty of the authors work.*

This is a good point. The manuscript clearly needs to do a better job of citing and discussing previous work such as Exbrayat et al. (2013a, 2013b), as well as more recent work (Guenet et al., 2018; Luo et al., 2016). In our revision we will provide more background on pattern scaling, as well as the state of CMIP5 models' carbon cycle performance more generally (Anav et al., 2013; Luo et al., 2016), and using aspects of model behavior to draw inferences about climate- and carbon-cycle response to anthropogenic forcing (e.g. Gillett et al., 2013). In addition, we think that a better discussion of how RH pattern scaling can be treated as a type of emergent constraint (Hoffman et al., 2014; Luo et al., 2015) would be useful.
* * *
*The second paragraph of the introduction states most models simulate increasing RH and that existing RH process representation is simple (first order kinetics) compared to many others ecosystem processes (I assume e.g. photosynthesis?). Then moving on to compare observation driven estimate of RH with NPP estimates. I think this is too many concepts in one paragraph without adequately describing any of them. Paragraph two should deal with a description of exiting RH model structures. Highlighting known issues with first order kinetic, e.g. lack of a microbial pool and difficulties in responding to changes in litter quality versus quantity (e.g. Wieder et al., 2013, Xenakis & Williams 2014). The importance of soil moisture (Exbrayat et al., 2013ab, Exbrayat et al., 2014) or nutrient cycles (Manzoni & Porporato 2009; Exbrayat et al., 2013a).*

This proposed structure ties in nicely with the previous comment, and would make things clearer for the reader, we agree.
* * *
*P1 L44-45 "While both temperature and precipitation have a positive effect on the global terrestrial carbon flux" Are you still talking about respiration? Net ecosystem exchange, Net biome exchange?*

This will be clarified.
* * *
*Methods:*
*Linking back to the decomposition review from the introduction details of which temperature and soil moisture response functions used in the CMIP5 models seems appropriate to me.*

This is a good idea, and will help the reader understand the links between these processes, their implementation in models, and how this links to the current study.

*P3 L1-6: Does the observation-driven estimates come with an uncertainty analysis?*

Unfortunately not really, no. We will discuss this issue and note frequent sources of error in these and other upscaled observational products (Hashimoto et al., 2015; Jung et al., 2011; Xiao et al., 2011).

*P3 L21: "...we only used the first realisation..." Would it not be more appropriate to use the mean across ensembles?*

This was also raised by Reviewer 4. Preliminary analyses suggested that there was little ensemble-to-ensemble variation for these variables in CMIP5, and we can include this as supplementary information; if the reviewers feel strongly about this point, however, we are happy to re-do the analysis using the mean of all ensemble members.

*Results:*
*P5 L14-20: It is not immediately clear whether you are talking about correlations in space or time. Also please be clear throughout the manuscript that precipitation is a proxy for soil moisture availability. Therefore you should not be talking about both soil moisture and rainfall being limiting. Soil moisture / plant available water is limited.*

Good point–thank you. We will clearly separate temporal and spatial correlations, as well as note the precipitation/soil moisture point.

*P5 L28-30: "This is likely due to less land (and this higher variability is model averages)..." Is it not equally or more likely that greater divergence between models occurs because the models are trained and developed using observations which are bias to the temperate northern hemisphere?*

Yes, and this ties in with a point we (somewhat obliquely) made on page 5 (see response to Reviewer 1). This revised manuscript will more clearly explain the potential links between inconsistent model performance in the 'global South' and lack of observations in those regions. We will also note that this problem has been addressed in other contexts, e.g. upscaling of FLUXNET data (Jung et al., 2011).

*P6 L13-21: Is there no observation equivalent for this analysis?*

We can do so, for example by discussing differing regional sensitivities to climate change (Liu et al., 2016; Schimel et al., 2015).

*P7 L1-14: I think both of these paragraphs need to be clearer. It is not always obvious to me whether you are talking about simulated vs observed RH. Would be improved if you make better use of the available figures in this section.*

We agree; currently the text too frequently is unclear in which specific result is being focused on
* * *
*Discussion:*
*P8 L19-20: This is the first time pools have been mentioned. This should be first mentioned in the introduction.*
*Moreover, I feel a more thorough discussion of the associated literature is needed.*

Yes, we agree, the introduction should–and will–do a better job of linking fluxes and stocks, and discussing their relationship. As noted above, the manuscript clearly needs to do a better job of citing and discussing previous work.
* * *
*P8 L25-46: I think both these paragraphs need rephrasing to improve clarity. There is no use of any figures or tables from you manuscript here.*

See previous comment; we agree, these paragraphs to be clearer about their links to specific results.
* * *
*P9 L15-17: Your text appears to be referring to the global average but what about spatial patterns? You present a large number of figures with spatial variation, can you make greater use of these?*

This is a good suggestion–thank you. We will look at this point and try to break the global means down, taking advantage of the spatial information presented.
* * *
*P9 L22-30: I think this would be a good area to discuss some of the possible parameterisation / model processes missing within the existing models within the context of the material I suggested should be added to the introduction.*

As noted above, we agree that this would significantly strengthen the context and interpretation surrounding our results.
* * *
*P9 L39: "...temperature response (Q10)..." please provide range for context.*

An interesting suggestion! We will do so.
* * *
*P1 L22 "Compared to observations, ESMs consistency..." -> "Compared to observations ESMs consistently..."*
*P1 L36 "carbon cycle" -> "carbon (C) cycle"*
*P2 L31 "...an observation-based data product." -> "...observation-driven analysis."*
*P2 L46 "(Hashimoto et al., 2015):(Hashimoto et al., 2015)"*
*P4 L40: "...majority (65 %)..." please state number of models.*
*P5 L1: "...smallest projected trend..." trend in what? not clear from text. P6 L24: "...weaker correlated models..."*

These points will be clarified and/or fixed.
* * *
*P7 L21: "...ESMs examined here (Figure X)" ?*
*P7 L23: delete "On one hand, "*
*P7 L25: delete "On the other,"*
*P7 L30: "...soil moisture response functions."*
*P7 L35: "robustly" I'm not convinced you can say this. "Consistently" would be a more appropriate word*
*P8 L1: delete "Interestingly,"*
*P8 L9: "...across empirical datasets." Such as?*
*P8 L25: "Because..." you should not begin a sentence with "because", let alone a paragraph or subsection. Please rephrase.*

We will fix these issues.

**References cited in response**

Anav, A., Friedlingstein, P., Kidston, M., Bopp, L., Ciais, P., Cox, P. M., Jones, C. D., Jung, M., Myneni, R. B. and Zhu, Z.: Evaluating the land and ocean components of the global carbon cycle in the CMIP5 earth system models, J. Clim., 26, 6801–6843, 2013.

Bond-Lamberty, B. and Thomson, A. M.: Temperature-associated increases in the global soil respiration record, Nature, 464(7288), 579–582, 2010.

Bond-Lamberty, B., Epron, D., Harden, J. W., Harmon, M. E., Hoffman, F. M., Kumar, J., McGuire, A. D. and Vargas, R.: Estimating heterotrophic respiration at large scales: challenges, approaches, and next steps, Ecosphere, 7(6), d01380, 2016.

Davidson, E. A. and Janssens, I. A.: Temperature sensitivity of soil carbon decomposition and feedbacks to climate change, Nature, 440, 165–173, 2006.

Exbrayat, J.-F., Pitman, A. J., Zhang, Q., Abramowitz, G. and Wang, Y.-P.: Examining soil carbon uncertainty in a global model: response of microbial decomposition to temperature, moisture and nutrient limitation, Biogeosciences, 10, 7095–7108, 2013a.

Exbrayat, J.-F., Pitman, A. J., Abramowitz, G. and Wang, Y.-P.: Sensitivity of net ecosystem exchange and heterotrophic respiration to parameterization uncertainty, J. Geophys. Res., doi:10.1029/2012JD018122, 2013b.

Falloon, P. D., Jones, C. D., Ades, M. and Paul, K. I.: Direct soil moisture controls of future global soil carbon changes: An important source of uncertainty, Global Biochemical Cycles, 25, GB3010, 2011.

Gillett, N. P., Arora, V. K., Matthews, D. and Allen, M. R.: Constraining the Ratio of Global Warming to Cumulative CO2 Emissions Using CMIP5 Simulations, J. Clim., 26(18), 6844–6858, 2013.

Guenet, B., Camino-Serrano, M., Ciais, P., Tifafi, M., Maignan, F., Soong, J. L. and Janssens, I. A.: Impact of priming on global soil carbon stocks, Glob. Chang. Biol., doi:10.1111/gcb.14069, 2018.

Hashimoto, S., Carvalhais, N., Ito, A., Migliavacca, M., Nishina, K. and Reichstein, M.: Global spatiotemporal distribution of soil respiration modeled using a global database, Biogeosciences, 12, 4121–4132, 2015.

Herger, N., Sanderson, B. W. and Knutti, R.: Improved pattern scaling approaches for the use in climate impact studies, Geophys. Res. Lett., in press, doi:10.1002/2015GL063569, 2015.

Hoffman, F. M., Randerson, J. T., Arora, V., Bao, Q., Six, K. D., Cadule, P., Ji, D., Jones, C. D., Kawamiya, M., Khatiwala, S., Lindsay, K., Obata, A., Shevliakova, E., Tjiputra, J., Volodin, E. M. and Wu, T.: Causes and implications of persistent atmospheric carbon dioxide biases in Earth System Models, Journal of Geophysical Research-Biogeosciences, in press, doi:10.1002/2013JG002381, 2014.

Hursh, A., Ballantyne, A., Cooper, L., Maneta, M., Kimball, J. and Watts, J.: The sensitivity of

soil respiration to soil temperature, moisture, and carbon supply at the global scale, Glob. Chang. Biol., 23(5), 2090–2103, 2017.

Jung, M., Reichstein, M., Margolis, H. A., Cescatti, A., Richardson, A. D., Arain, M. A., Arneth, A., Bernhofer, C., Bonal, D., Chen, J., Gianelle, D., Gobron, N., Kiely, G., Kutsch, W. L., Lasslop, G., Law, B. E., Lindroth, A., Merbold, L., Montagnani, L., Moors, E. J., Papale, D., Sottocornola, M., Vaccari, F. and Williams, C. A.: Global patterns of land-atmosphere fluxes of carbon dioxide, latent heat, and sensible heat derived from eddy covariance, satellite, and meteorological observations, Journal of Geophysical Research-Biogeosciences, 116, G00J07, 2011.

Jung, M., Reichstein, M., Schwalm, C. R., Huntingford, C., Sitch, S., Ahlström, A., Arneth, A., Camps-Valls, G., Ciais, P., Friedlingstein, P., Gans, F., Ichii, K., Jain, A. K., Kato, E., Papale, D., Poulter, B., Raduly, B., Rödenbeck, C., Tramontana, G., Viovy, N., Wang, Y.-P., Weber, U., Zaehle, S. and Zeng, N.: Compensatory water effects link yearly global land CO2 sink changes to temperature, Nature, 541(7638), 516–520, 2017.

Le Quéré, C., Andrew, R. M., Friedlingstein, P., Sitch, S., Pongratz, J., Manning, A. C., Korsbakken, J. I., Peters, G. P., Canadell, J. G., Jackson, R. B., Boden, T. A., Tans, P. P., Andrews, O. D., Arora, V. K., Bakker, D. C. E., Barbero, L., Becker, M., Betts, R. A., Bopp, L., Chevallier, F., Chini, L. P., Ciais, P., Cosca, C. E., Cross, J., Currie, K., Gasser, T., Harris, I., Hauck, J., Haverd, V., Houghton, R. A., Hunt, C. W., Hurtt, G., Ilyina, T., Jain, A. K., Kato, E., Kautz, M., Keeling, R. F., Klein Goldewijk, K., Körtzinger, A., Landschützer, P., Lefèvre, N., Lenton, A., Lienert, S., Lima, I., Lombardozzi, D., Metzl, N., Millero, F., Monteiro, P. M. S., Munro, D. R., Nabel, J. E. M. S., Nakaoka, S.-I., Nojiri, Y., Padín, X. A., Peregon, A., Pfeil, B., Pierrot, D., Poulter, B., Rehder, G., Reimer, J., Rödenbeck, C., Schwinger, J., Séférian, R., Skjelvan, I., Stocker, B. D., Tian, H., Tilbrook, B., van der Laan-Luijkx, I. T., van der Werf, G. R., van Heuven, S., Viovy, N., Vuichard, N., Walker, A. P., Watson, A. J., Wiltshire, A. J., Zaehle, S. and Zhu, D.: Global Carbon Budget 2017, Earth Syst. Sci. Data Discuss., 1–79, 2017.

Link, R., Bond-Lamberty, B., Hartin, C., Lynch, C. and Kravitz, B.: Computationally efficient emulators for Earth System Models, Geoscientific Model Development, submitted, 2018.

Liu, L., Wang, X., Lajeunesse, M. J., Miao, G., Piao, S., Wan, S., Wu, Y., Wang, Z., Yang, S. and Deng, M.: A cross-biome synthesis of soil respiration and its determinants under simulated precipitation changes, Glob. Chang. Biol., 22(4), 1394–1405, 2016.

Luo, Y., Keenan, T. F. and Smith, M.: Predictability of the terrestrial carbon cycle, Glob. Chang. Biol., 21(5), 1737–1751, 2015.

Luo, Y., Ahlström, A., Allison, S. D., Batjes, N. H., Brovkin, V., Carvalhais, N., Chappell, A., Ciais, P., Davidson, E. A., Finzi, A., Georgiou, K., Guenet, B., Hararuk, O., Harden, J. W., He, Y., Hopkins, F., Jiang, L., Koven, C., Jackson, R. B., Jones, C. D., Lara, M. J., Liang, J., McGuire, A. D., Parton, W., Peng, C., Randerson, J. T., Salazar, A., Sierra, C. A., Smith, M. J., Tian, H., Todd-Brown, K. E. O., Torn, M., van Groenigen, K. J., Wang, Y. P., West, T. O., Wei, Y., Wieder, W. R., Xia, J., Xu, X., Xu, X. and Zhou, T.: Toward more realistic projections of soil carbon dynamics by Earth system models, Global Biogeochem. Cycles, 30(1), 2015GB005239, 2016.

Mahecha, M. D., Reichstein, M., Carvalhais, N., Lasslop, G., Lange, H., Seneviratne, S. I., Vargas, R., Ammann, C., Arain, M. A., Cescatti, A., Janssens, I. A., Migliavacca, M.,

Montagnani, L. and Richardson, A. D.: Global convergence in the temperature sensitivity of respiration at ecosystem level, Science, 329(5993), 838–840, 2010.

McGuire, A. D., Anderson, L. G., Christensen, T. R., Dallimore, S., Guo, L., Hayes, D. J., Heimann, M., Lorenson, T. D., Macdonald, R. W. and Roulet, N. T.: Sensitivity of the carbon cycle in the Arctic to climate change, Ecol. Monogr., 79(4), 523–555, 2009.

Mitchell, T. D.: Pattern Scaling: An Examination of the Accuracy of the Technique for Describing Future Climates, Clim. Change, 60(3), 217–242, 2003.

Moyano, F. E., Manzoni, S. and Chenu, C.: Responses of soil heterotrophic respiration to moisture availability: An exploration of processes and models, Soil Biol. Biochem., 59, 72–85, 2013.

Odum, E. P.: The strategy of ecosystem development, Science, 164, 262–270, 1969.

Phillips, C. L., Bond-Lamberty, B., Desai, A. R., Lavoie, M., Risk, D., Tang, J., Todd-Brown, K. and Vargas, R.: The value of soil respiration measurements for interpreting and modeling terrestrial carbon cycling, Plant Soil, 413(1-2), 1–25, 2017.

Rehfeld, K., Münch, T., Ho, S. L. and Laepple, T.: Global patterns of declining temperature variability from the Last Glacial Maximum to the Holocene, Nature, 554(7692), 356–359, 2018.

Schimel, D., Stephens, B. B. and Fisher, J. B.: Effect of increasing CO2 on the terrestrial carbon cycle, Proceedings of the National Academy of Science, 112(2), 436–441, 2015.

Shao, P., Zeng, X., Moore, D. J. P. and Zeng, X.: Soil microbial respiration from observations and Earth System Models, Environ. Res. Lett., 8(3), 034034, 2013.

Tebaldi, C. and Arblaster, J. M.: Pattern scaling: Its strengths and limitations, and an update on the latest model simulations, Clim. Change, 122(3), 459–471, 2014.

Todd-Brown, K. E. O., Randerson, J. T., Post, W. M., Hoffman, F. M., Tarnocai, C., Schuur, E. A. G. and Allison, S. D.: Causes of variation in soil carbon predictions from CMIP5 Earth system models and comparison with observations, Biogeosciences, 10, 1717–1736, 2013.

Xiao, J., Davis, K. J., Urban, N. M., Keller, K. and Saliendra, N. Z.: Upscaling carbon fluxes from towers to the regional scale: Influence of parameter variability and land cover representation on regional flux estimates, Journal of Geophysical Research-Biogeosciences, 116, GB3027, 2011.

---

## Author Comment (AC4) · 23 Feb 2018

Please see responses in supplement attached.

Please also note the supplement to this comment:
https://www.biogeosciences-discuss.net/bg-2017-405/bg-2017-405-AC4-supplement.pdf
* * *